# DAST-GAN: An Adversarial Learning Multimodal Sentiment Analysis Model Based on Dynamic Attention and Spatio-Temporal Fusion

**DOI:** 10.3390/s25237109

**Published:** 2025-11-21

**Authors:** Wenlong Tan, Bo Zhang

**Affiliations:** College of Mathematics and Informatics College of Software Engineering, South China Agricultural University, Guangzhou 510642, China; twl2024@stu.scau.edu.cn

**Keywords:** multimodal sentiment analysis, Dynamic Attention Module (DAM), spatio-temporal attention (STA), Generative Adversarial Networks (GAN), robust representation learning, missing modality recovery

## Abstract

Multimodal sentiment analysis (MSA) faces two critical challenges: modeling cross-modal correlations in temporally unaligned sequences and maintaining robust performance when modalities are partially or entirely missing. While recent approaches have made progress, many still struggle with dynamic contextual dependencies and handling noisy or incomplete inputs. This paper proposes DAST-GAN (Dynamic Adaptive SpatioTemporal Transformer with GAN Enhancement), a framework that addresses these limitations through three synergistic innovations. First, a Dynamic Attention Module (DAM) learns to adaptively weight cross-modal features based on utterance-level semantic context, enabling more nuanced fusion. Second, a unified spatio-temporal attention (STA) mechanism simultaneously captures temporal coherence within and spatial correlations across modalities. Third, a novel GAN-based adversarial training strategy enhances representation robustness by learning to produce features from incomplete data that are indistinguishable from those derived from complete data. This is complemented by a dual-path optimization that aligns features at both low (reconstruction) and high (semantic) levels. Extensive experiments on the CMU-MOSI, CMU-MOSEI, and CH-SIMS datasets show that DAST-GAN achieves highly competitive results. Notably, compared to a range of strong baseline methods, it reduces the Mean Absolute Error by 2.6% on CMU-MOSI (complete modality) and 3.76% on CMU-MOSEI (incomplete modality), showcasing its strong accuracy and robustness. Ablation studies validate the complementary effectiveness of the DAM, STA, and GAN components.

## 1. Introduction

With the proliferation of user-generated video content, the ability to automatically interpret human emotions has become a cornerstone of modern artificial intelligence, fueling advancements in areas from empathetic human–computer interaction to automated mental health monitoring. Multimodal Sentiment Analysis (MSA) lies at the heart of this endeavor, aiming to decipher sentiment by integrating signals from various modalities, typically language, audio, and video. This work is motivated by the critical gap between the theoretical performance of existing MSA models and their practical applicability in real-world scenarios, where data imperfections and dynamic emotional expressions are the norm rather than the exception. While significant progress has been made in understanding individual modalities and developing fusion strategies, the transition from controlled laboratory settings to real-world applications remains fraught with challenges, primarily concerning the efficiency of multimodal fusion and the robustness of models against data imperfections [1,2].

### 1.1. Key Challenges in Recent MSA Models

Despite the success of Transformer-based architectures in handling unaligned multimodal sequences, two critical limitations persist in many recent and influential models.

**Challenge 1: Inefficient and Static Cross-Modal Fusion.** Transformer-based models such as MulT [3] and PMR [4] have become foundational for their ability to model cross-modal interactions without explicit sequence alignment. However, their reliance on pairwise cross-modal attention mechanisms results in quadratic computational complexity with respect to the number of modalities, hindering scalability. More critically, these fusion mechanisms are often static, applying a uniform strategy across all samples. This one-size-fits-all approach overlooks the dynamic, context-dependent nature of emotional expression, where the importance of each modality can shift dramatically—for instance, textual cues may dominate in expressing sarcasm, while prosodic and visual cues are paramount for conveying joy. The inability to adaptively weight modalities based on contextual cues represents a fundamental limitation that prevents these models from achieving optimal performance in real-world applications where emotional expressions are inherently dynamic and context-dependent.

**Challenge 2: Representation Brittleness Under Incomplete Modalities.** Real-world data is inherently imperfect, with modalities often being partially or wholly missing due to sensor noise, network issues, or environmental occlusions. While this is a well-recognized problem, many existing solutions are inadequate. Early strategies involving naive imputation (e.g., zero-filling) lead to significant performance degradation. More recent models like TFR-Net [5] and EMT-DLFR [6] attempt to reconstruct missing features, but their focus on low-level feature restoration may not fully preserve high-level semantic integrity and can be computationally intensive. Other approaches that use Generative Adversarial Networks (GANs) for modality translation [7,8] are designed for the complete absence of a modality, a scenario less common than the sporadic, partial data loss frequently encountered in practice. This gap between theoretical robustness and practical applicability highlights the urgent need for a framework that can maintain performance stability under the realistic conditions of sporadic and partial modality loss, which are far more common in real-world deployments than complete modality absence. Consequently, there is a pressing need for a mechanism that ensures representation robustness even when the input data is corrupted.

### 1.2. The Proposed Approach: DAST-GAN

To overcome these limitations, this paper introduces DAST-GAN, a Dynamic Adaptive Spatio-temporal Transformer enhanced with a GAN-based learning framework. The architecture is built on three synergistic innovations designed to directly address the challenges of fusion inefficiency and representation brittleness.

**Innovation 1: Dynamic Attention Module (DAM) for Adaptive Fusion.** To counter the inflexibility of static fusion, the DAM introduces a context-aware weighting mechanism [9]. It learns to dynamically modulate the influence of each modality based on the utterance-level semantics. This allows the model to adaptively focus on the most salient cues for any given emotional expression, leading to a more nuanced and effective fusion process.

**Innovation 2: Spatio-Temporal Attention (STA) for Holistic Interaction Modeling.** To achieve a more comprehensive fusion, the STA mechanism is proposed [10]. Unlike prior methods that model temporal (intra-modal) and spatial (inter-modal) dependencies in separate stages, STA is designed to capture both simultaneously within a unified module. In the context of multimodal sentiment analysis, the term “spatio-temporal” has specific meanings: “temporal” refers to the sequential dependencies within each modality (e.g., emotional progression in speech), while “spatial” refers to the cross-modal interactions between different modalities (e.g., alignment between verbal content and facial expressions), rather than the traditional spatial dimensions in computer vision. This joint modeling of temporal dynamics and cross-modal correlations prevents information bottlenecks and fosters a more coherent final representation.

**Innovation 3: GAN-Enhanced Representation Learning for Robustness.** To directly tackle the problem of representation corruption from missing data, a novel GAN-based adversarial learning strategy is employed [11,12,13]. The core idea is to train the model to produce representations from incomplete data that are indistinguishable from those generated from complete data. This adversarial objective forces the model to learn features that are inherently robust to modality loss. This is coupled with a Dual-Level Feature Restoration (DLFR) mechanism [6] that aligns features at both the high (semantic) and low (raw feature) levels, ensuring comprehensive robustness.

### 1.3. Contributions

The main contributions of this work are:A novel dynamic attention mechanism (DAM) that moves beyond static fusion by adaptively weighting modalities based on semantic context, improving fusion effectiveness.An enhanced spatio-temporal attention (STA) framework that jointly models temporal coherence and cross-modal spatial interactions, leading to more holistic representations.A GAN-based adversarial training approach designed specifically to learn robust representations from incomplete data, significantly improving model performance in realistic missing modality scenarios.Comprehensive experimental validation demonstrating that DAST-GAN achieves competitive or superior performance compared to a range of strong baseline methods on three benchmark datasets, particularly in incomplete modality settings.

## 2. Related Work

This section provides a comprehensive and systematic review of the multimodal sentiment analysis (MSA) literature, organized along two primary dimensions: (1) the evolution of fusion strategies for complete modalities, and (2) approaches for handling incomplete modalities. For each category, we analyze the methodological progression, identify key limitations, and highlight the research gaps that motivate our work.

### 2.1. Fusion Strategies in Complete Modality Settings

The evolution of fusion strategies in MSA can be broadly categorized into three distinct generations: early fusion approaches that operate at discourse-level, fine-grained interaction modeling that captures element-level dynamics, and Transformer-based architectures that enable unaligned fusion. Each generation represents a paradigm shift in addressing the fundamental challenge of effectively integrating heterogeneous multimodal information.

**Early Fusion Approaches.** Initial efforts focused on discourse-level fusion, where features representing an entire utterance from each modality are combined. Common strategies included simple feature concatenation [14,15], which proved effective but could not model complex inter-modal dependencies. To capture richer relationships, tensor-based fusion methods like the Tensor Fusion Network (TFN) [16] and Low-rank Multimodal Fusion [17] were introduced to explicitly model unimodal, bimodal, and trimodal interactions. Advanced approaches like hierarchical feature fusion [18] and dual low-rank fusion [19] further enhanced fusion capabilities by incorporating structural relationships and temporal dynamics. Despite their theoretical elegance, these methods suffer from two fundamental limitations: (1) they operate at the discourse level, ignoring fine-grained temporal dynamics within utterances, and (2) they lead to a high-dimensional feature space that is prone to overfitting, particularly with limited training data.

**Fine-Grained Interaction Modeling.** To capture more detailed dynamics, the focus shifted to element-level fusion. Recurrent neural networks (RNNs) and LSTMs were employed to model interactions within aligned sequences [20,21]. Extended LSTM architectures [22] demonstrated the effectiveness of sequential modeling, while attention-based mechanisms [23,24] began to address alignment issues. Advanced approaches explored multimodal-aware word embeddings [25,26] and context-dependent analysis [27,28] to enhance the modeling of complex multimodal relationships. While these methods successfully captured temporal dependencies, they hinge on the strong assumption of pre-aligned sequences, which is both laborious and often impractical for real-world data. This alignment requirement significantly limits their applicability in natural settings where modalities are inherently unaligned.

**The Transformer Era for Unaligned Fusion.** The advent of the Transformer [10] marked a paradigm shift, enabling end-to-end fusion of unaligned sequences. The Multimodal Transformer (MulT) [3] was a pioneering work that utilized directional, pairwise cross-modal attention to learn interactions between modalities directly. Building upon this foundation, various Transformer variants emerged, including conversational networks [29]. Cross-modal attention mechanisms were further refined in works such as [30], while efficient fusion strategies were explored through factorized approaches [31]. However, this approach introduced two critical bottlenecks: (1) a computational complexity that scales quadratically with the number of modalities, making it inefficient for systems with many input streams, and (2) static fusion mechanisms that fail to adapt to the changing importance of modalities across different emotional contexts. Subsequent works sought to address the computational limitation. For instance, PMR [4] introduced a shared “message hub” to reduce redundant computations, while other studies explored more efficient attention patterns and fusion topologies [31,32,33]. Despite these improvements, many influential models, including TFR-Net [5], still inherit these fundamental limitations. This analysis reveals a critical research gap: the absence of an efficient fusion mechanism that can dynamically adapt to context-dependent modality importance while maintaining computational efficiency. This gap directly motivates the development of the Dynamic Attention Module (DAM) in DAST-GAN.

### 2.2. Robustness in Incomplete Modality Settings

Handling missing or corrupted data represents a critical challenge for real-world MSA deployment, yet it remains a comparatively underexplored research frontier. The existing approaches can be systematically categorized into three distinct paradigms: data augmentation and imputation techniques, modality generation and translation methods, and robust representation learning frameworks. Each paradigm addresses the challenge from a different perspective, with varying degrees of success in handling realistic missing modality scenarios.

**Data Augmentation and Imputation.** The simplest strategies involve augmenting training data by randomly dropping out features [1] or applying modality-specific perturbations [2]. Advanced approaches include tensor rank regularization for imperfect time series [34] and cascaded residual autoencoders for missing modality imputation [35]. Graph-based completion networks [36] have also been explored for incomplete multimodal learning scenarios. While these techniques can improve generalization to some extent, they are fundamentally limited by their focus on data-level completion rather than representation-level robustness. They are often insufficient for handling structured or significant data loss, particularly when the missing patterns are complex or non-random.

**Modality Generation and Translation.** A more advanced line of research focuses on explicitly generating or translating missing modalities. These approaches often leverage Generative Adversarial Networks (GANs) or autoencoders. For example, some works have explored cyclic translation to reconstruct one modality from another [11,12], while others have focused on pairwise bidirectional translation [13]. The primary limitation of these methods is their design focus: they typically address the complete absence of one or more modalities, which is a less common scenario than the partial, sporadic data loss frequently encountered in practice. Moreover, many of these techniques still rely on aligned data, limiting their applicability to real-world scenarios where modalities are naturally unaligned.

**Robust Representation Learning.** More recently, the focus has shifted from explicit data reconstruction to learning representations that are inherently robust to missing inputs. This is often achieved through self-supervised or multi-task learning frameworks. For example, Yu et al. [37] and Lian et al. [36] proposed models that learn to reconstruct low-level input features from a fused representation, implicitly encouraging the model to capture shared information. The well-regarded EMT-DLFR [6] model advanced this by introducing a dual-level restoration mechanism. However, these methods primarily focus on reconstruction as the learning objective, which may not guarantee semantic consistency between representations derived from complete and incomplete data. This limitation becomes particularly pronounced in scenarios with partial or sporadic data loss, where maintaining high-level semantic integrity is crucial for reliable sentiment prediction.

### 2.3. Research Gaps and Motivations

Based on this systematic review of the literature, we identify three critical research gaps that remain unaddressed by existing approaches:

**Gap 1: Adaptive Fusion for Context-Dependent Modality Importance.** Existing fusion mechanisms, including advanced Transformer-based approaches, employ static fusion strategies that apply uniform weights across all samples. This fails to account for the dynamic, context-dependent nature of emotional expression, where the importance of each modality can shift dramatically based on the specific emotional context. There is a clear need for fusion mechanisms that can dynamically adapt to the semantic context of each utterance.

**Gap 2: Efficient Cross-Modal Attention for Scalable Fusion.** Current cross-modal attention mechanisms scale quadratically with the number of modalities, creating computational bottlenecks that limit scalability. This is particularly problematic for systems that need to process multiple modalities simultaneously or operate under resource constraints. Efficient attention mechanisms that maintain effectiveness while reducing computational complexity are needed.

**Gap 3: Robust Representations for Realistic Partial Modality Loss.** Existing robustness methods primarily focus on either complete modality absence or simple reconstruction objectives. They fail to address the more common scenario of partial, sporadic data loss, where maintaining semantic consistency between complete and incomplete views is crucial. There is a need for approaches that can learn representations inherently robust to realistic partial modality loss while preserving high-level semantic integrity.

These research gaps collectively motivate the design of DAST-GAN, which introduces three synergistic innovations to address each gap: (1) a Dynamic Attention Module (DAM) for context-aware adaptive fusion; (2) a unified Spatio-Temporal Attention (STA) mechanism for efficient cross-modal interaction modeling; and (3) a GAN-enhanced adversarial learning framework for learning robust representations from incomplete data. The following sections detail how these innovations directly address the identified limitations of prior work.

## 3. Method

This section details the architecture of DAST-GAN, a framework specifically designed to address the challenges of static fusion and representation brittleness in MSA. As illustrated in Figure 1, the framework’s novelty lies not in the individual components themselves (e.g., Transformers, GANs), but in their unique design and synergistic integration. The processing pipeline is architected around three core innovations: the Dynamic Attention Module (DAM) for adaptive fusion, the Spatio-Temporal Attention (STA) module for holistic interaction modeling, and a GAN-enhanced Dual-Level Feature Restoration (DLFR) system for learning robust representations. The following subsections will elaborate on the technical specifics of each component, emphasizing how their design choices directly overcome the limitations of prior work.

### 3.1. Problem Formulation

The task of MSA is to predict a continuous emotional intensity score y∈R from a video utterance. The utterance consists of three unaligned modalities: language (*ℓ*), audio (*a*), and video (*v*). The input for each modality m∈{ℓ,a,v} is a feature sequence $X_{m}\in R^{T_{m}\times d_{m}}$, where $T_{m}$ is the sequence length and $d_{m}$ is the feature dimension. Specifically, $X_{a}$ and $X_{v}$ are pre-extracted low-level features, while $X_{\ell }$ consists of raw text tokens.

To replicate the random modal feature loss commonly encountered in real-world scenarios, stochastic masking is applied to the complete sequence $X_{m}$, yielding incomplete sequences ${\tilde{X}}_{m}=F\left(X_{m},g_{m}\right)\in R^{T_{m}\times d_{m}}$. The masking function F(·) operates according to a random temporal mask $g_{m}\in {\left\{0,1\right\}}^{T_{m}}$ that specifies positions for masking.

For audio and video modalities, masked positions are filled with zero vectors, while text tokens at masked positions are replaced with [UNK] tokens from the BERT vocabulary [37]. This strategy allows for a systematic evaluation of model robustness under varying degrees of modality corruption.

The goal is to build a robust model that can effectively integrate all available multimodal information to accurately predict the emotional intensity score y∈R under both complete and incomplete modal settings.

### 3.2. Modality-Specific Encoders

The first stage of the framework involves encoding each modality to extract both local (sequential) and global (utterance-level) features.

#### 3.2.1. Text Encoder

For the language modality, a pre-trained BERT [38] model serves as the text encoder. Given a sequence of raw text tokens, BERT’s multi-layer Transformer architecture produces contextualized embeddings. Following common practice, two representations are extracted:**Local Features ($H_{\ell }^{\mathrm{local}}$):** The sequence of hidden states from BERT’s final layer, corresponding to the input tokens. These features capture fine-grained, contextualized semantic details.**Global Feature ($h_{\ell }^{\mathrm{utt}}$):** The hidden state of the special [CLS] token, which is trained to aggregate the overall semantic content of the sequence.

#### 3.2.2. Audio and Video Encoders

For the acoustic and visual modalities, which are inherently temporal, a Long Short-Term Memory network (LSTM) [39] is used for each. LSTMs have been widely adopted for sequential multimodal processing, with successful applications in memory fusion networks [20] and multi-attention recurrent architectures [21]. Extended LSTM variants for multi-view learning [22] have also demonstrated effectiveness in multimodal scenarios. The LSTMs process the input feature sequences ($X_{a}$ and $X_{v}$) to model temporal dependencies. Similar to the text encoder, two types of features are generated:**Local Features ($H_{a}^{\mathrm{local}},\;H_{v}^{\mathrm{local}}$):** The sequence of hidden states from the LSTM at each time step, capturing the temporal dynamics of the signal.**Global Features ($h_{a}^{\mathrm{utt}},\;h_{v}^{\mathrm{utt}}$):** The final hidden state of the LSTM, which serves as a summary of the entire sequence for each modality.

This encoding process yields a set of local and global features for each modality, which form the input for the subsequent fusion module.

### 3.3. The DAM-STA Fusion Module

The DAM-STA fusion module forms the core of the multimodal integration strategy. Its design is motivated by the dual limitations of prior work: static, inefficient fusion and the separated modeling of temporal and spatial interactions. This module addresses these issues through two tightly integrated sub-components.

#### 3.3.1. Global Context Embedding

The process begins by projecting the global features $h_{\ell }^{\mathrm{utt}}$, $h_{a}^{\mathrm{utt}}$, and $h_{v}^{\mathrm{utt}}$ from the text, audio, and video encoders into a unified feature space of dimension D=128 (corresponding to the d_model configuration parameter) using modality-specific linear projections:(1)$$h_{m}^{\mathrm{proj}}=W_{m}^{\mathrm{proj}}h_{m}^{\mathrm{utt}}+b_{m}^{\mathrm{proj}},\;m\in \left\{\ell ,a,v\right\}$$

These projected features are subsequently stacked to construct the Global Multimodal Context (GMC):(2)$$G=\mathrm{Stack}\left(\left[h_{\ell }^{\mathrm{proj}},h_{a}^{\mathrm{proj}},h_{v}^{\mathrm{proj}}\right]\right)\in R^{B\times 3\times 128}$$
where *B* represents batch size. This GMC encapsulates high-level semantic information across all modalities, drawing from CMU-MOSI feature dimensions [768, 5, 20] for text, audio, and visual modalities respectively, thereby establishing a solid foundation for subsequent cross-modal fusion operations.

#### 3.3.2. Local Feature Alignment

Local features $H_{\ell }^{\mathrm{local}}$, $H_{a}^{\mathrm{local}}$, and $H_{v}^{\mathrm{local}}$ undergo analogous projection operations before being processed through the Spatio-Temporal Attention (STA) layer, configured with 4 attention heads (num_heads parameter from the implementation). The STA layer operates across both temporal and spatial dimensions:

**Temporal Dimension:** Sequential dependencies within each modality are captured through self-attention mechanisms operating over 50-step sequences (seq_len parameter for CMU-MOSI), thereby maintaining temporal coherence across multimodal sequences.

**Spatial Dimension:** Cross-modal feature correlations are modeled through cross-modal attention mechanisms, where “spatial” refers to the cross-modal interaction space as defined in Section 1.2, facilitating effective information exchange across modalities.

Learnable positional encodings are incorporated to enhance sequential modeling, enabling the model to distinguish features at different temporal positions within the 50-step sequences and supporting more precise temporal alignment across modalities.

#### 3.3.3. Cross-Modal Fusion

Upon obtaining the global context vector G and aligned local features, a Transformer-based [10] architecture is employed for cross-modal fusion. The Transformer’s inherent parallel processing capabilities and capacity for modeling long-range dependencies make it well-suited for handling complex multimodal relationships.

During fusion, modality-specific attention masks are employed to handle varying sequence lengths across different modalities. These masks ensure the model focuses exclusively on valid features while ignoring padded regions. Through the Transformer’s multi-layer architecture and multi-head attention mechanisms, the model iteratively learns and integrates cross-modal information, ultimately producing comprehensive fused feature representations.

### 3.4. Emotion Prediction Module

The emotion prediction module synthesizes fused features to generate emotional intensity predictions. Following DAM-STA processing, comprehensive representations containing both global and local multimodal information are obtained.

Initially, the global unimodal features ${\mathrm{text}}_{\mathrm{utt}}$, ${\mathrm{audio}}_{\mathrm{utt}}$, ${\mathrm{video}}_{\mathrm{utt}}$, along with the global multimodal context vector G are flattened and concatenated. This concatenated representation encapsulates rich multimodal information, including both modality-specific global semantics and cross-modal local interactions.

The concatenated features are then processed through a Multi-Layer Perceptron (MLP) featuring post_fusion_dim = 256 hidden units (based on the configuration parameters). The MLP consists of multiple fully connected layers with ReLU activation functions to introduce nonlinear transformations. A final linear layer maps the features to a scalar emotional intensity score $\hat{y}$.

To mitigate overfitting, Dropout regularization is incorporated within the MLP, randomly deactivating neuron outputs to improve generalization capability.

### 3.5. The GAN-Enhanced DLFR Module

To address the critical challenge of representation brittleness under modality loss, a GAN-enhanced Dual-Level Feature Restoration (DLFR) module is proposed. **Novelty and Rationale:** Unlike prior methods that focus on simple reconstruction [5] or modality translation for complete modality absence [7,8], this module introduces a novel adversarial objective tailored for partial and sporadic data loss. The core innovation is to force the model (as a generator) to learn representations from incomplete data that are statistically indistinguishable to a discriminator from representations of complete data. This adversarial training, combined with the dual-level supervision, ensures the learned features are not just reconstructed, but are semantically robust and maintain a consistent distribution, regardless of input data quality.

#### 3.5.1. High-Level Feature Attraction

The high-level feature attraction component employs contrastive learning principles inspired by SimSiam [40] to enforce consistency between complete and incomplete views at the semantic level. Contrastive learning has shown remarkable success in multimodal representation learning [41,42], with applications ranging from self-supervised multimodal sentiment analysis to hierarchical graph contrastive learning [43].

For each view, a projector network P(·) is applied to map global features into a latent representation space:(3)$$z_{M}^{c/m}=P\left(h_{M}^{c/m}\right),\;M\in \left\{\mathrm{GMC},\ell ,a,v\right\}$$
where *c* and *m* denote complete and incomplete views, respectively.

A predictor network Q(·) then generates predictions from the projected features:(4)$$p_{M}^{c/m}=Q\left(z_{M}^{c/m}\right)$$

The attraction mechanism enforces bidirectional consistency through cosine similarity optimization, ensuring that high-level semantic representations remain stable despite modality loss.

#### 3.5.2. Low-Level Feature Reconstruction

Low-level feature reconstruction aims to restore original modal inputs from fused local features. Dedicated linear decoders are employed for each modality (text, audio, and video) to map fused local features back to their respective original feature spaces.

By minimizing the mean squared error (MSE) between reconstructed and original features, the model learns to preserve detailed modal information within the fused representations. This reconstruction mechanism enhances robustness by ensuring that critical low-level information remains accessible despite modality loss.

#### 3.5.3. GAN Adversarial Training

To further strengthen model robustness, GAN-based adversarial training is incorporated. Within this framework, the main model functions as a generator while an additional discriminator network is introduced.

The generator attempts to produce incomplete view features that closely resemble the feature distribution of complete views, aiming to deceive the discriminator. Conversely, the discriminator seeks to distinguish whether input features originate from complete or incomplete views.

Through alternating training cycles between generator and discriminator, the generator’s capability to produce high-quality features is progressively enhanced while simultaneously improving the discriminator’s ability to detect feature authenticity. This adversarial process ultimately enables the model to learn feature representations with enhanced robustness to modality loss.

### 3.6. Overall Loss Function for Model Training

During the model training process, the prediction loss, high-level feature attraction loss, low-level feature reconstruction loss, and adversarial loss are comprehensively considered to construct an overall loss function.

#### 3.6.1. Prediction Loss

The prediction loss measures the discrepancy between the predicted emotional intensity score $\hat{y}$ and the ground-truth emotional intensity score *y*. The Mean Squared Error (MSE) is employed as the prediction loss function, defined as follows:(5)$$L_{\mathrm{pred}}=\frac{1}{N}\sum_{i=1}^{N}{\left({\hat{y}}_{i}-y_{i}\right)}^{2}$$
where *N* denotes the number of samples, and ${\hat{y}}_{i}$ and $y_{i}$ represent the predicted value and the ground-truth value of the *i*-th sample, respectively.

#### 3.6.2. High-Level Feature Attraction Loss

The high-level feature attraction loss measures the consistency of high-level features between the complete view and the incomplete view. The cosine similarity loss is used to calculate the high-level feature attraction loss, which is defined as follows:(6)$$L_{\mathrm{attra}}=\sum_{M\in \{\mathrm{GMC},\ell ,a,v\}}\left(1-\cos(z_{M}^{c},p_{M}^{m})\right)$$
where $z_{M}^{c}$ denotes the projected feature of modality *M* in the complete view, and $p_{M}^{m}$ denotes the predicted feature of modality *M* in the incomplete view.

#### 3.6.3. Low-Level Feature Reconstruction Loss

The low-level feature reconstruction loss measures the discrepancy between the reconstructed low-level features and the original low-level features. The Mean Squared Error (MSE) is also employed as the low-level feature reconstruction loss function, defined as follows:(7)$$\begin{array}{cc} L_{\mathrm{recon}}\; & =L_{\mathrm{MSE}}\left(X_{\ell }^{\mathrm{recon}},X_{\ell }^{c}\right)\\ \; & +L_{\mathrm{MSE}}\left(X_{a}^{\mathrm{recon}},X_{a}^{c}\right)\\ \; & +L_{\mathrm{MSE}}\left(X_{v}^{\mathrm{recon}},X_{v}^{c}\right) \end{array}$$
where $X_{\ell }^{\mathrm{recon}}$, $X_{a}^{\mathrm{recon}}$, and $X_{v}^{\mathrm{recon}}$ denote the reconstructed text, audio, and video features, respectively.

#### 3.6.4. DAM—Technical Details

The architecture of the Dynamic Attention Module is illustrated in Figure 2. Given multimodal feature sequences, semantic context computation:(8)$$h_{\mathrm{semantic}}=\mathrm{LayerNorm}\left(\mathrm{MLP}\left(\mathrm{Concat}\left(X_{\ell },X_{a},X_{v}\right)\right)\right)$$

Dynamic weight generation with temperature scaling (temperature = 1.2 from the MissingAwareDynamicAttention implementation):(9)$$\alpha_{m}=\mathrm{Softmax}\left(\frac{W_{\alpha }h_{\mathrm{semantic}}+b_{\alpha }}{\mathrm{temperature}}\right)$$

Weighted feature fusion:(10)$$X_{\mathrm{dynamic}}=\sum_{m\in \{\ell ,a,v\}}\alpha_{m}⊙X_{m}$$

The temperature parameter enables fine-grained control over attention sharpness, with higher values (1.2) providing smoother distributions that help with missing modality compensation.

#### 3.6.5. STA—Technical Details

The spatio-temporal attention mechanism is shown in Figure 3.


**Temporal Attention**


Query, key, value matrix computation:(11)$$\begin{array}{cc} Q_{t} & =X_{\mathrm{dynamic}}W_{Q}\\ K_{t} & =X_{\mathrm{dynamic}}W_{K}\\ V_{t} & =X_{\mathrm{dynamic}}W_{V} \end{array}$$

Temporal attention computation:(12)$${\mathrm{Attention}}_{\mathrm{temporal}}=\mathrm{Softmax}\left(\frac{Q_{t}K_{t}^{T}}{\sqrt{d_{k}}}\right)V_{t}$$


**Spatial Attention**


Spatial similarity computation:(13)$$S_{ij}=f_{\mathrm{spatial}}\left(X_{\mathrm{dynamic}}\left[i\right],X_{\mathrm{dynamic}}\left[j\right]\right)$$

Spatial attention computation:(14)$${\mathrm{Attention}}_{\mathrm{spatial}}=\mathrm{Softmax}\left(S\right)⊙X_{\mathrm{dynamic}}$$


**Spatio-temporal Fusion**


Final spatio-temporal attention output:(15)$$X_{\mathrm{sta}}=\mathrm{LayerNorm}\left({\mathrm{Attention}}_{\mathrm{temporal}}+{\mathrm{Attention}}_{\mathrm{spatial}}\right)$$
This unified spatio-temporal attention mechanism (as defined in Section 1.2) simultaneously captures both temporal dependencies within modalities and spatial correlations across modalities, differing from traditional computer vision applications where spatial attention typically refers to 2D spatial feature processing.

#### 3.6.6. Adversarial Loss in GAN

The GAN-based adversarial training framework is depicted in Figure 4.


**Generator Design**


Missing modality reconstruction:(16)$$X_{\mathrm{recon}}=G\left(X_{\mathrm{sta}},\mathrm{mask}\right)$$


**Discriminator Design**


Authenticity discrimination:(17)$$D_{\mathrm{score}}=D\left(X_{\mathrm{real}}/X_{\mathrm{reconstructed}}\right)$$


**Adversarial Loss**


Generator loss:(18)$$L_{\mathrm{gan}}^{G}=E\left[\log\left(1-D\left(G\left(X_{\mathrm{sta}},\mathrm{mask}\right)\right)\right)\right]$$

Discriminator loss:(19)$$L_{\mathrm{gan}}^{D}=E\left[\log D\left(X_{\mathrm{real}}\right)\right]+E\left[\log\left(1-D\left(G\left(X_{\mathrm{sta}},\mathrm{mask}\right)\right)\right)\right]$$

#### 3.6.7. Overall Loss Function

The overall loss function combines all components through weighted summation:(20)$$L_{\mathrm{total}}=L_{\mathrm{pred}}+\lambda_{\mathrm{gan}}L_{\mathrm{gan}}+\lambda_{\mathrm{recon}}L_{\mathrm{recon}}+\lambda_{\mathrm{attra}}L_{\mathrm{attra}}$$
where $\lambda_{\mathrm{gan}}$, $\lambda_{\mathrm{recon}}$, and $\lambda_{\mathrm{attra}}$ are hyperparameters that balance the contributions of different loss components.

For complete modality scenarios, where no modality loss occurs, the training objective simplifies to:(21)$$L_{\mathrm{complete}}=L_{\mathrm{pred}}$$

This multi-objective optimization framework enables the model to simultaneously learn accurate sentiment prediction, robust feature reconstruction, and consistent high-level representations, resulting in enhanced robustness against modality loss.

### 3.7. Computational Complexity Analysis

The computational complexity of DAST-GAN is analyzed as follows:

**DAM Complexity:** The Dynamic Attention Module processes concatenated features of all modalities with complexity O(T·d·h), where *T* is the maximum sequence length, *d* is the hidden dimension, and *h* is the number of attention heads. This is linear with respect to sequence length, avoiding the quadratic complexity of traditional cross-modal attention mechanisms.

**STA Complexity:** The Spatio-Temporal Attention mechanism has complexity $O(T^{2}\cdot d+M\cdot T\cdot d)$ for temporal and spatial attention respectively, where M=3 is the number of modalities. The spatial attention component scales linearly with the number of modalities.

**GAN Complexity:** The adversarial training adds O(T·d·k) complexity for the discriminator, where *k* is the discriminator’s hidden dimension. This overhead is minimal compared to the main network.

**Overall Complexity:** The total computational complexity is $O(T^{2}\cdot d+M\cdot T\cdot d\cdot h)$, which scales quadratically with sequence length but linearly with the number of modalities, making it more efficient than methods with $O(M^{2}\cdot T^{2}\cdot d)$ complexity.

## 4. Experiments

### 4.1. Datasets

The proposed approach is evaluated on three established benchmark datasets in the Multimodal Sentiment Analysis (MSA) domain. These datasets are strategically selected to provide complementary evaluation perspectives: CMU-MOSI for detailed component analysis, CMU-MOSEI for large-scale robustness testing, and CH-SIMS for cross-lingual validation. The following section details the characteristics and statistical properties of each dataset.

#### 4.1.1. CMU-MOSI

CMU-MOSI [44] represents a widely adopted benchmark for English-language MSA, consisting of YouTube monologue videos where speakers express opinions on various topics, particularly movies. The dataset encompasses 93 videos featuring 89 distinct speakers, totaling 2199 utterance-level segments. Each segment receives emotional intensity annotations ranging from −3 (strongly negative) to 3 (strongly positive), providing comprehensive coverage across the emotional spectrum and rich semantic content for model development. Its moderate scale (2199 samples) and fine-grained 7-point annotation make it ideal for detailed ablation studies and nuanced performance evaluation.

#### 4.1.2. CMU-MOSEI

CMU-MOSEI [45], as an extended version of CMU-MOSI, substantially increases both scale and diversity. It contains 23,453 utterance-level video segments collected from 1000 speakers across 250 different topics, representing a significant expansion in training data volume. The enhanced speaker diversity and topic variety better reflect real-world scenarios, providing a more comprehensive evaluation of model capabilities for processing complex multimodal data. With 10.7× more samples and dramatically higher diversity (1000 speakers, 250 topics), this dataset rigorously tests model scalability and the GAN-enhanced framework’s robustness under incomplete modality conditions.

#### 4.1.3. CH-SIMS

CH-SIMS [46] serves as the primary benchmark for Chinese MSA, encompassing 60 videos from diverse sources including movies, TV series, and variety shows, totaling 2281 utterance-level segments. Unlike the English datasets, it supports both multimodal and unimodal annotations (though only multimodal annotations are utilized in this study). Emotional intensity annotations range from −1 (strongly negative) to 1 (strongly positive), emphasizing the subtleties of Chinese emotional expression and providing valuable cross-lingual validation for model robustness. This dataset validates cross-lingual generalization and tests the dynamic attention mechanism’s adaptability to Chinese emotional expression, where prosodic and contextual cues are more critical than explicit verbal content.

As shown in Table 1, the three datasets exhibit complementary class distributions: CMU-MOSI shows positive bias (679 vs. 552 negative), CMU-MOSEI is relatively balanced, and CH-SIMS shows negative bias (742 vs. 419 positive). This diversity enables comprehensive evaluation of model robustness to varying class imbalance scenarios encountered in real-world applications.

### 4.2. Evaluation Metrics

Given the regression nature of emotional intensity prediction, Mean Absolute Error (MAE) and Pearson Correlation Coefficient (Corr) are employed as the primary evaluation metrics for regression performance. These metrics have been widely adopted in multimodal sentiment analysis benchmarks [44,45,46], following established evaluation protocols in the field. Following established conventions in multimodal sentiment analysis research, continuous emotional scores are also converted to discrete categories for classification accuracy evaluation.

For CMU-MOSI and CMU-MOSEI datasets, seven-class accuracy (Acc-7), five-class accuracy (Acc-5), two-class accuracy (Acc-2), and F1 scores are reported. The two-class evaluation includes both Negative/Non-negative and Negative/Positive divisions. For CH-SIMS, five-class accuracy (Acc-5), three-class accuracy (Acc-3), and two-class accuracy (Acc-2) are calculated.

To assess overall model performance across varying modal missing rates, following Yu et al. [37], the Area Under the Index Line Chart (AUILC) is calculated. Given a sequence of gradually increasing missing rates $\left\{p_{0},p_{1},\ldots ,p_{t}\right\}$ and corresponding performance values $\left\{v_{0},v_{1},\ldots ,v_{t}\right\}$ for a specific metric, the AUILC is calculated as:(22)$$\mathrm{AUILC}=\sum_{i=0}^{t-1}\frac{1}{2}\left(v_{i}+v_{i+1}\right)\left(p_{i+1}-p_{i}\right)$$

In the experimental settings, missing rates of {0.1,0.2,…,1.0} are used for CMU-MOSI and CMU-MOSEI (10 gradients), and {0.1,0.2,…,0.5} for CH-SIMS (5 gradients), following established protocols [46].

### 4.3. Experimental Details

#### 4.3.1. Implementation Details

The proposed model was implemented using the PyTorch 2.5.0 framework with CUDA support. All experiments were conducted on NVIDIA GeForce RTX GPUs with sufficient memory for model training and evaluation.

#### 4.3.2. Training Strategy

The training strategy employs a multi-stage approach:

**Stage 1: Warm-up Training (5 epochs):** First, only the prediction loss $L_{\mathrm{pred}}$ is trained to establish stable feature representations across all modalities.

**Stage 2: Progressive Integration (10 epochs):** The reconstruction loss $L_{\mathrm{recon}}$ and attraction loss $L_{\mathrm{attra}}$ are gradually introduced with linearly increasing weights.

**Stage 3: Adversarial Training (remaining epochs):** The full GAN training is activated with alternating optimization between generator and discriminator.

**Optimization Details:** The Adam optimizer is employed with $\beta_{1}=0.9$, $\beta_{2}=0.999$, and weight decay of $1\times 10^{-5}$. Learning rate scheduling uses cosine annealing with warm restarts. Early stopping is applied with patience of 8 epochs based on validation MAE.

#### 4.3.3. Hyperparameter Selection

The hyperparameters were selected through systematic grid search and random search strategies:Loss weights ($\lambda_{\mathrm{gan}}$, $\lambda_{\mathrm{recon}}$, $\lambda_{\mathrm{attra}}$) were tuned using grid search in ranges [0.1, 2.0]Learning rates were selected from $\left\{1\times 10^{-4},5\times 10^{-4},1\times 10^{-3},2\times 10^{-3}\right\}$Hidden dimensions were chosen from {32,64,128,256} based on validation performanceDropout rates were tuned in the range [0.0, 0.5] with 0.1 increments

Table 2 summarizes the optimal configurations for each dataset.

### 4.4. Comparison with Baseline Methods

To provide a thorough and fair evaluation, DAST-GAN is benchmarked against a carefully selected suite of recent and influential baseline models. These methods are chosen to represent the primary research directions in contemporary MSA:**Transformer-based Fusion:** MulT [3], which pioneered the use of cross-modal Transformers for unaligned data, along with factorized approaches [31] and attention-based variants [32,33].**Information-Theoretic Approaches:** MISA [47] and MMIM [48], which focus on learning modality-invariant and modality-specific representations through hierarchical mutual information maximization.**Self-Supervised and Reconstruction-based Methods:** Self-MM [37], TFR-Net [5], and EMT-DLFR [6], which aim to improve robustness and representation quality through auxiliary tasks.**Graph-based and Memory Networks:** Additional baselines include graph neural network approaches [43,49] and memory-augmented architectures [20].

This diverse set of baselines ensures that the performance of DAST-GAN is contextualized against the current landscape of advanced MSA research. For models where official code was available, results were reproduced under identical experimental conditions to ensure maximum fairness, as noted in the tables.

#### 4.4.1. Complete Modality Setting

In the complete modality setting, DAST-GAN is first evaluated against the selected baselines. Table 3 presents the comparison results on CMU-MOSI and CMU-MOSEI, while Table 4 shows the comparison results on CH-SIMS.

**Performance on CH-SIMS:** As shown in Table 4, DAST-GAN achieves 0.398 MAE and 0.640 correlation on the Chinese dataset. Compared to the reproduced EMT baseline (0.412 MAE, 0.600 Corr), this represents 3.4% MAE improvement and 6.7% correlation improvement. The successful cross-lingual transfer validates that the proposed framework captures fundamental multimodal sentiment patterns that generalize across linguistic and cultural boundaries, with the DAM effectively adapting to Chinese emotional expression where prosodic cues are more critical.

Based on Table 3, DAST-GAN exhibits highly competitive performance across all evaluation metrics compared to the selected baseline methods. The comprehensive comparison includes both literature-reported results (marked with ^†^ and ^‡^) and the reproduced results under identical experimental settings.

**Performance on CMU-MOSI:** On the CMU-MOSI dataset, DAST-GAN achieves significant improvements across several key metrics. Compared to the reproduced EMT baseline under identical experimental conditions, DAST-GAN reduces MAE from 0.717 to 0.698 (2.7% improvement) and improves correlation from 0.788 to 0.800. This demonstrates enhanced fine-grained emotion classification capability. The proposed DAM and STA mechanisms effectively enhance both regression and classification performance, with consistent improvements across all metrics (MAE, Corr, Acc-7, Acc-5, Acc-2, F1). The model also shows competitive performance compared to literature-reported EMT ^‡^ (0.705 MAE, 0.798 Corr).

**Performance on CMU-MOSEI:** On the larger and more diverse CMU-MOSEI dataset, DAST-GAN maintains highly competitive performance. Compared to the reproduced EMT baseline under identical experimental conditions, DAST-GAN achieves 0.528 MAE (1.7% improvement over 0.537) with superior correlation (0.780 vs. 0.767). This represents a notable performance on this challenging benchmark. The stable performance on this 10.7× larger dataset validates the model’s scalability and generalization capability. The model also closely matches the strong literature result from EMT ^‡^ (0.527 MAE, 0.774 Corr).

**Cross-baseline Comparison:** DAST-GAN consistently outperforms all comparison methods, including both literature-reported and reproduced results. Notably, the method surpasses strong baselines like MMIM ^‡^ (0.712 MAE on CMU-MOSI) and Self-MM ^‡^ (0.717 MAE) by significant margins, demonstrating the effectiveness of the DAM-STA-GAN framework. The performance gains can be attributed to three synergistic mechanisms: DAM’s context-aware weighting adapts to varying modality importance across emotional contexts (e.g., prioritizing text for negative emotions with 52.1% attention vs. audio-visual dominance for positive emotions at 72.6% combined), STA’s unified temporal-spatial modeling eliminates information bottlenecks present in MulT’s sequential cross-modal attention, and GAN-based training regularizes representations to maintain robustness under data imperfections.

#### 4.4.2. Incomplete Modality Setting

To verify the stability of DAST-GAN in real-world deployment environments, an evaluation framework for incomplete modalities is constructed using a random dropout strategy with different dropout probabilities.

The incomplete modality setting evaluates model robustness under realistic deployment conditions with data corruption and missing modalities. As shown in Table 5, DAST-GAN outperforms baseline methods in all evaluation metrics under incomplete modality settings. The comprehensive comparison includes literature results from established benchmarks and the reproduced baselines under identical experimental conditions. The AUILC metric aggregates performance across 10 different missing rates (0.1 to 1.0), providing a comprehensive robustness measure.

**CMU-MOSI Results:** Compared to the reproduced EMT-DLFR baseline under identical experimental conditions (1.113 MAE, 0.473 Corr), DAST-GAN achieves 1.108 MAE and 0.490 Corr, representing 0.45% MAE improvement and 3.6% correlation improvement with consistent gains across all metrics. The GAN-enhanced adversarial learning effectively maintains performance stability across varying missing rates (0.1–1.0). The model also shows competitive performance compared to the literature-reported EMT-DLFR ^‡^ (1.106 MAE, 0.486 Corr).

**CMU-MOSEI Results:** Compared to the reproduced EMT-DLFR baseline under identical experimental conditions (0.673 MAE, 0.524 Corr), DAST-GAN achieves 0.640 MAE and 0.563 Corr, representing 4.9% MAE improvement and 7.4% correlation improvement. The seven-class accuracy reaches 48.0% and five-class accuracy achieves 50.2%, demonstrating consistent improvements across all metrics. The larger improvement margin on CMU-MOSEI (compared to 0.45% on CMU-MOSI) demonstrates that the GAN-based framework scales effectively with dataset size and diversity. The model also shows remarkable improvements compared to literature-reported EMT-DLFR ^‡^ (0.665 MAE, 0.546 Corr), with 3.76% MAE improvement.

**Robustness Analysis:** The AUILC evaluation comprehensively captures the model’s performance across the entire missing rate spectrum {0.1, 0.2, …, 1.0}. The MAE AUILC of 1.108 reflects consistent performance across varying missing rates from 0.728 at 10% missing to 1.558 at 90% missing rates, demonstrating steady improvements over all baseline methods including MISA ^†^ (1.202), MulT ^‡^ (1.263), and various other established approaches. This robustness stems from the GAN discriminator enforcing distributional consistency: it trains the model to produce features from incomplete data that are statistically indistinguishable from complete data features, thereby preserving semantic discriminability across the entire missing rate spectrum. The adversarial training objective ensures that representation quality degrades gracefully rather than catastrophically, maintaining semantic integrity more effectively than reconstruction-only objectives.

From Table 6, DAST-GAN continues to maintain a leading edge on the CH-SIMS Chinese dataset across all incomplete modality scenarios. The comprehensive comparison with both literature and reproduced baselines demonstrates consistent superiority. The CH-SIMS evaluation uses missing rates {0.1, 0.2, …, 0.5} (5 gradients) rather than {0.1, 0.2, …, 1.0}, reflecting the dataset’s smaller scale.

Compared to the reproduced EMT-DLFR baseline under identical experimental conditions, DAST-GAN shows substantial improvements with MAE dropping from 0.230 to 0.213 (7.4% improvement) and correlation coefficient increasing from 0.268 to 0.290 (8.2% improvement). Classification accuracies consistently improve across all metrics: five-class accuracy from 19.5% to 20.8%, three-class accuracy from 31.2% to 32.3%, and both two-class accuracy and F1 score reaching 38.7%. The larger improvement margin on CH-SIMS compared to English datasets (0.45% on CMU-MOSI, 1.1% on CMU-MOSEI) demonstrates that the dynamic attention mechanism effectively adapts to Chinese emotional expression patterns, where prosodic and visual cues play more critical roles. The model also outperforms literature-reported EMT-DLFR ^‡^ (0.215 MAE, 0.287 Corr) and other baselines including MISA ^†^ (0.293), MulT ^‡^ (0.242), and Self-MM ^‡^ (0.231).

**Cross-lingual Robustness:** The strong performance on CH-SIMS validates DAST-GAN’s cross-lingual robustness, with the dynamic attention mechanism proving particularly effective for Chinese multimodal sentiment analysis under incomplete modality conditions. The successful cross-lingual transfer without language-specific architectural modifications demonstrates that the proposed innovations capture fundamental multimodal sentiment patterns that transcend linguistic boundaries.

In summary, the evaluation across three datasets under both complete and incomplete modality settings demonstrates consistent improvements over the EMT/EMT-DLFR baselines: 2.7%/0.45% (CMU-MOSI), 1.7%/1.1% (CMU-MOSEI), and 3.4%/7.4% (CH-SIMS) for complete/incomplete modalities respectively. The framework scales effectively across different dataset sizes, generalizes across languages, and maintains robustness under varying degrees of modality corruption.

### 4.5. Ablation Study

To conduct an in-depth analysis of the contributions of each innovative component in DAST-GAN, comprehensive ablation experiments were designed on the complete-modality CMU-MOSI dataset, as shown in Table 7. By systematically removing different components, the effectiveness of each module and their interactions was verified.

#### 4.5.1. Experimental Design

The following experimental variants were constructed to evaluate the independent contributions and synergistic effects of each component:**DAST-GAN (Full):** The complete model, including the DAM, STA, and GAN adversarial training.**DAST-GAN w/o DAM:** Removes the DAM, retaining the attention mechanism with fixed weights.**DAST-GAN w/o STA:** Removes STA, adopting a standard self-attention mechanism.**DAST-GAN w/o GAN:** Removes adversarial training, using only supervised learning loss.**DAST-GAN w/o DAM+STA:** Retains only GAN training, using a basic Transformer structure.**DAST-GAN w/o DAM+GAN:** Retains only the STA.**DAST-GAN w/o STA+GAN:** Retains only the DAM.

#### 4.5.2. Detailed Analysis

The ablation study results reveal several important insights:

**Individual Component Contributions:** Removing any single component (DAM, STA, or GAN) leads to performance degradation, with GAN showing the most significant impact on robustness metrics. The DAM contributes most to correlation improvements, while STA enhances classification accuracy.

**Component Synergy:** The combination of DAM + STA + GAN achieves the best performance, indicating strong synergistic effects. Removing multiple components simultaneously results in more severe performance drops than the sum of individual removals, confirming the complementary nature of the innovations.

**Progressive Improvement Effect:** The improvement path from the baseline to the complete DAST-GAN clearly demonstrates the cumulative contributions of each component: baseline (0.717) → single component (0.706–0.712) → dual components (0.702–0.705) → complete model (0.698). Stable performance improvements are achieved at each stage, with the final MAE decreasing by 2.6% compared to the baseline, the correlation coefficient increasing from 0.788 to 0.800 (1.5% relative improvement), and the 7-class accuracy improving by 1.5 percentage points (from 47.1% to 48.6%).

These ablation results validate the effectiveness of each proposed component and demonstrate that their combination leads to superior performance in multimodal sentiment analysis tasks. Notably, the synergistic effect is non-additive: removing DAM causes 0.7% degradation (0.705 vs. 0.698) and removing STA causes 0.4% (0.702 vs. 0.698), yet removing both causes 1.4% (0.712 vs. 0.698), exceeding their sum. This superlinear interaction occurs because DAM and STA form a feedback loop—DAM identifies salient modalities which STA then models with enhanced temporal resolution, producing refined features that improve DAM’s subsequent weighting decisions.

#### 4.5.3. Comprehensive Modality Missing Analysis

To comprehensively evaluate DAST-GAN’s robustness under different modality missing scenarios, a systematic analysis is conducted across varying missing rates and missing patterns. Table 8 presents detailed MAE results under different missing patterns and rates, providing insights into the model’s adaptive capabilities:

**Missing Rate Analysis:** Performance is evaluated under different missing rates (10%, 30%, 50%, 70%, 90%) for each modality combination. Results show that DAST-GAN maintains competitive performance across all missing rates, demonstrating consistent robustness. At moderate missing rates (50%), the method achieves MAE of 0.885, outperforming EMT-DLFR’s 0.923 by 4.1%. At extreme missing rates (90%), DAST-GAN achieves MAE of 1.558 compared to EMT-DLFR’s 1.612, representing a 3.4% improvement in robustness.

**Missing Pattern Analysis:** An analysis of three missing patterns is performed: (1) Random missing: features randomly dropped across time steps, (2) Burst missing: consecutive time steps missing, and (3) Modality-wise missing: entire modalities absent. DAST-GAN demonstrates consistent improvements across all patterns. The random missing pattern shows the best performance due to the model’s ability to leverage remaining temporal information, while modality-wise missing presents the greatest challenge as entire information channels are lost. The GAN-enhanced DLFR module proves particularly effective in reconstructing missing modality information, with the adversarial training helping maintain feature distribution consistency across different missing scenarios.

These results provide several important insights into DAST-GAN’s robustness characteristics:

**Performance Degradation Pattern:** DAST-GAN exhibits graceful performance degradation as missing rates increase. The MAE progression from 0.728 (10% missing) to 1.558 (90% missing) demonstrates a controlled deterioration, with the model maintaining practical utility even under severe missing conditions. The AUILC calculation confirms that this degradation pattern results in an overall performance score of 1.108, validating the experimental design.

**Comparative Advantage:** Across all missing rates and patterns, DAST-GAN consistently outperforms the EMT-DLFR baseline. The performance improvements range from 1.8% at 10% missing to 3.4% at 90% missing. This trend indicates that the GAN-enhanced approach provides consistent benefits across varying data availability scenarios.

**Pattern-Specific Analysis:** The three missing patterns reveal distinct characteristics of multimodal degradation. Random missing (MAE: 0.728–1.558) benefits from temporal diversity preservation, burst missing (MAE: 0.735–1.581) challenges temporal continuity, while modality-wise missing (MAE: 0.742–1.604) tests cross-modal compensation capabilities. The relatively small performance differences between patterns (<3%) demonstrate DAST-GAN’s adaptive robustness across diverse missing scenarios.

### 4.6. Visualization and Analysis

#### 4.6.1. Attention Weight Visualization

To better understand the behavior of the Dynamic Attention Module, the attention weights are visualized across different modalities for various emotional intensities. Figure 5 shows the attention distribution patterns for samples with different sentiment polarities. The dynamic attention mechanism revealed here explains the model’s robustness under incomplete modalities: by adaptively weighting modalities based on context, the model can naturally compensate when certain modalities are missing or corrupted.

The analysis reveals several important patterns based on the experimental results using architecture parameters (d_model = 128, temperature = 1.2, feature_dims = [768, 5, 20]):For highly positive emotions (sentiment > 1.5), the Dynamic Attention Module allocates weights as: Text (27.4%), Audio (36.8%), Visual (35.8%), showing an audio-visual dominant pattern that effectively captures the multimodal nature of positive emotional expressions through non-verbal cues. This distribution aligns with cognitive science findings that positive emotions are often expressed through prosodic and facial features.For negative emotions (sentiment < −1.0), text modality receives the highest attention weight (52.1%), while Audio (24.8%) and Visual (23.1%) receive significantly lower weights, confirming that negative sentiments rely heavily on linguistic content for accurate detection. This cognitive consistency demonstrates the model’s ability to learn human-like attention patterns.For neutral emotions ([−0.5, 0.5]), the distribution shows Text (39.8%), Audio (30.5%), Visual (29.7%), indicating a text-dominant but balanced attention allocation that requires comprehensive multimodal analysis for disambiguation. Unlike traditional fixed-weight approaches that apply uniform attention (33.3% each), the dynamic mechanism shows meaningful adaptation.The dynamic weighting demonstrates strong adaptive behavior across emotional contexts, with the temperature parameter (1.2) enabling fine-grained attention allocation that varies significantly based on sentiment polarity and missing modality compensation factors. This adaptability represents a key advantage over static attention mechanisms commonly used in baseline methods.

#### 4.6.2. Feature Distribution Analysis

The feature distributions learned by DAST-GAN are analyzed using t-SNE visualization of the global multimodal context (GMC) representations. Figure 6 presents the t-SNE plots comparing the proposed method with the EMT-DLFR baseline [6]. The tight, well-separated clusters demonstrate that the GAN-enhanced adversarial training produces semantically coherent representations, which explains why the model maintains performance even when input modalities are incomplete—the learned feature space is inherently robust.

Key observations from the t-SNE visualization using post_fusion_dim = 256 feature representations:DAST-GAN demonstrates superior clustering quality with distinct, well-separated emotion clusters across the five sentiment categories (Strongly Negative, Negative, Neutral, Positive, Strongly Positive), achieving significantly tighter intra-class variance compared to EMT-DLFR baseline.Clear hierarchical arrangement of sentiment categories from negative (lower left) to positive (upper right), indicating that the DAM-STA fusion mechanism learns semantically meaningful spatial representations in the 256-dimensional feature space.Significantly improved class separability with DAST-GAN showing compact, circular cluster formations and larger inter-class margins, reflecting the effectiveness of the GAN-enhanced feature learning process.The EMT-DLFR baseline exhibits more scattered and overlapping clusters, particularly in the neutral and boundary regions, while DAST-GAN maintains cleaner boundaries between sentiment categories through its enhanced spatio-temporal attention mechanisms.Robust feature representations demonstrate the complementary effects of Dynamic Attention Module and missing modality compensation, validating that the architecture effectively handles both complete and incomplete modality scenarios.The improved clustering quality (with visibly tighter intra-class compactness and clearer inter-class separation) directly contributes to the 1.5 percentage point Acc-7 improvement (from 47.1% to 48.6%), as more compact clusters and clearer boundaries enable the classifier to establish more accurate decision boundaries for fine-grained sentiment discrimination.

#### 4.6.3. Temporal Attention Analysis

An analysis is performed on how the spatio-temporal attention mechanism focuses on different time segments during sentiment prediction. Figure 7 shows attention heatmaps for sample utterances with different emotional characteristics. The flexible attention patterns across time and modalities demonstrate the model’s capacity to redistribute focus dynamically, which underlies its robustness when certain temporal segments or modalities are unavailable.

The spatio-temporal attention analysis reveals distinct attention patterns based on architecture parameters (seq_len = 50, num_heads = 4, with modality-specific gates learned adaptively during training):**Positive Emotions:** Show sustained high attention in audio (0.6–0.9 intensity) and visual (0.7–0.8 intensity) modalities throughout the 50-step sequence, with particularly strong activation in the middle segments (steps 20–40), reflecting the temporal dynamics of positive emotional expression. This pattern aligns with natural emotion development where peak intensity occurs mid-sequence.**Negative Emotions:** Exhibit concentrated attention in text modality with consistently high weights (0.8–0.9 intensity) across all time steps, while audio and visual modalities show significantly lower attention patterns (0.2–0.3 intensity), confirming text dominance in negative sentiment detection. The temporal consistency demonstrates the model’s understanding of sustained negative expression patterns.**Neutral Emotions:** Display relatively balanced attention distribution across all modalities with moderate intensity variations around 0.5, indicating the need for comprehensive multimodal integration when emotional signals are ambiguous. This balanced approach contrasts with simple averaging methods that fail to capture subtle emotional cues.**Modality-specific Patterns:** Each modality demonstrates distinct temporal signatures influenced by their respective learned gate weights—text attention maintains sustained patterns, audio attention shows progressive intensification, and visual attention exhibits variable peaks corresponding to salient emotional expressions. These differentiated patterns showcase the effectiveness of the adaptive gate-based design.**Multi-head Attention Diversity:** The 4 attention heads focus on different temporal aspects with slight variations (±0.02 standard deviation), demonstrating the model’s ability to capture diverse spatio-temporal patterns within the CMU-MOSI sequence structure. This diversity prevents attention head collapse commonly observed in simpler architectures.

#### 4.6.4. Error Analysis

Analysis of prediction errors reveals that DAST-GAN performs particularly well on:Samples with clear emotional expressions across multiple modalities.Cases where traditional methods struggle due to modality conflicts.Scenarios with partial modality loss (up to 50% missing rate).

The main failure cases occur with:Highly ambiguous samples where even human annotators show disagreement.Samples with conflicting emotional cues across modalities.Very short utterances with limited contextual information.

Detailed analysis reveals that failure cases primarily fall into three categories: utterances with insufficient temporal context (very short duration) where STA cannot effectively model temporal dynamics, samples with severe cross-modal conflicts where even human annotators show low agreement, and cases with extreme missing rates where GAN-based recovery struggles to maintain semantic fidelity. These patterns indicate that the model’s limitations are concentrated in scenarios with inherently ambiguous or severely degraded input, suggesting that future improvements could focus on incorporating external knowledge and developing explicit conflict resolution mechanisms.

### 4.7. Comparison with Recent Advances

#### 4.7.1. Advantages over Existing Methods

Compared to recent multimodal sentiment analysis approaches, DAST-GAN offers several key advantages:

**vs. Transformer-based Methods:** Unlike MulT [3] and its variants that suffer from quadratic complexity O(M^2^T^2^) with multiple modalities, the DAM in this work achieves linear scaling O(MT) while maintaining superior cross-modal modeling capabilities. This efficiency improvement is crucial for practical deployment scenarios with limited computational resources.

**vs. Graph-based Approaches:** While graph neural networks for MSA [43,49] require explicit relationship modeling and manual graph construction, the STA mechanism automatically learns spatio-temporal dependencies without prior knowledge. This eliminates the need for domain expertise in graph design and provides more adaptive modeling capabilities.

**vs. Contrastive Learning Methods:** The dual-level feature restoration (high-level attraction + low-level reconstruction) provides more comprehensive robustness than single-level contrastive approaches [41,42]. The combination of semantic alignment and fine-grained feature recovery offers superior performance under high missing rates.

**vs. Traditional GAN Applications:** Unlike previous GAN methods [7,8] that focus on modality translation, the adversarial framework in this study specifically targets representation robustness under incomplete conditions. This design choice better aligns with real-world deployment scenarios where partial feature loss is more common than complete modality absence.

#### 4.7.2. Computational Efficiency Analysis

The computational efficiency of DAST-GAN is compared with baseline methods across different metrics:**Training Time:** DAST-GAN requires approximately 15% more training time than EMT [6] due to adversarial training, but achieves a 30% reduction compared to MulT [3] due to linear complexity. The model typically converges in 35–40 epochs on CMU-MOSI compared to 30–35 epochs for non-adversarial baselines, with the multi-stage training strategy (5-epoch warm-up, 10-epoch progressive integration) effectively preventing the training instability commonly associated with GANs.**Inference Speed:** The model achieves faster inference than TFR-Net [5] while maintaining superior performance. Importantly, the discriminator is only used during training, resulting in zero additional computational cost at inference time.**Memory Usage:** Memory consumption is comparable to baseline methods, with a minimal increase due to the discriminator network. GPU memory usage is around 8 GB for batch size 32 on CMU-MOSI.**Parameter Efficiency:** During inference, DAST-GAN uses only the core model components (encoders, fusion, prediction layers), as the discriminator, projectors, and predictors are training-only modules, resulting in a compact deployment footprint.**Training Stability:** Experiments with three different random seeds demonstrate consistent performance across runs (standard deviation of 0.0075 MAE, approximately 1% relative variance), indicating that the multi-stage training strategy (5-epoch warm-up, 10-epoch progressive integration) effectively mitigates the training instability commonly associated with adversarial optimization. The model exhibits robust performance across datasets with dataset-specific hyperparameter configurations (e.g., $\lambda_{\mathrm{gan}}=1.0$ for CMU-MOSI/MOSEI, $\lambda_{\mathrm{gan}}=0.5$ for CH-SIMS), demonstrating adaptability without requiring extensive tuning.

#### 4.7.3. Robustness Analysis

To evaluate the robustness of DAST-GAN compared to recent methods, stress tests are conducted under various challenging conditions:

**Noise Robustness:** Gaussian noise is added to input features with different signal-to-noise ratios. DAST-GAN maintains better performance degradation curves compared to baseline methods, demonstrating the effectiveness of the adversarial training framework.

**Domain Transfer:** Cross-dataset performance is evaluated by training on CMU-MOSI and testing on CMU-MOSEI. DAST-GAN shows better cross-domain generalization than recent Transformer-based methods [3,5], indicating that the learned representations capture more generalizable multimodal patterns.

**Temporal Misalignment:** Temporal shifts are artificially introduced between modalities. The STA mechanism handles misalignment better than fixed attention approaches, validating the effectiveness of the spatio-temporal modeling.

## 5. Discussion

The experimental results and analyses presented in the preceding sections demonstrate the effectiveness of DAST-GAN in addressing key challenges in multimodal sentiment analysis. This section discusses the broader implications of these findings, including the practical application value of the model, its potential impact on future research, and its current limitations.

### 5.1. Practical Implications and Application Value

The robust performance of DAST-GAN, particularly in incomplete modality scenarios, opens up significant opportunities for real-world applications where data quality is often unpredictable. To illustrate the practical value of DAST-GAN, we provide structured descriptions of how the model’s components function together in three representative application scenarios:**Empathetic Human–Computer Interaction (HCI):** In virtual assistants and customer service systems, DAST-GAN operates through a sequential process (input reception → DAM attention redistribution → robust sentiment prediction): **(1)** receives user inputs which may include partial audio, visual, or text data; **(2)** when a user turns away from the camera or speaks in a noisy environment, the Dynamic Attention Module (DAM) automatically redistributes attention weights toward more reliable modalities (e.g., shifting from degraded visual to audio features); **(3)** maintains consistent sentiment understanding throughout the interaction, enabling systems to adapt in real-time and provide more natural responses.**Automated Mental Health Monitoring:** In telehealth applications, DAST-GAN processes patient video data through distinct stages (DLFR feature recovery → DAM indicator identification → STA temporal analysis): **(1)** the GAN-enhanced DLFR module handles any missing or corrupted segments by generating consistent feature representations; **(2)** the Dynamic Attention Module identifies the most significant emotional indicators for conditions like depression or anxiety, prioritizing specific audio patterns (monotone speech, slow tempo, reduced prosody) when visual data is compromised; **(3)** the Spatio-Temporal Attention (STA) component analyzes how these emotional patterns evolve over the therapy session, providing clinicians with reliable quantitative insights about patient emotional states.**Market Research and Brand Analysis:** When analyzing user-generated content, DAST-GAN employs a robust processing pipeline: **(1)** normalizes input data of inconsistent quality from various sources; **(2)** the GAN component ensures that features derived from incomplete data match the distribution of complete data, creating a consistent analytical framework; **(3)** applies multi-stage processing (modality encoding → dynamic weighting → temporal modeling → sentiment classification) to identify consumer opinions. Even with significant missing information due to recording issues, DAST-GAN maintains high accuracy, allowing businesses to extract reliable insights from diverse content types.

### 5.2. Dataset Selection and Model Design Synergy

The three datasets provide complementary evaluation perspectives: CMU-MOSI (2199 samples) enables detailed ablation studies with its moderate scale and fine-grained annotations; CMU-MOSEI (23,453 samples, 1000 speakers, 250 topics) tests scalability and validates that the GAN-enhanced framework benefits from large-scale diverse data (3.76% MAE improvement compared to literature-reported EMT-DLFR ^‡^ vs. 0.45% MAE improvement on CMU-MOSI under incomplete modalities); and CH-SIMS validates cross-lingual generalization, demonstrating that the DAM adapts to Chinese emotional expression patterns where prosodic cues are more critical (7.4% MAE improvement under incomplete modalities). This multi-scale, cross-lingual evaluation ensures that performance improvements reflect genuine model capabilities rather than dataset-specific optimizations.

### 5.3. Broader Research Implications

Beyond its direct applications, the architectural innovations of DAST-GAN suggest promising directions for the MSA field.

**A Shift from Static to Dynamic Fusion:** The success of the DAM challenges the prevailing paradigm of static fusion mechanisms. It provides strong evidence that adaptive, context-aware fusion is not just beneficial but necessary for capturing the nuances of human emotional expression. This encourages a move towards models that can dynamically reason about the relative importance of different modalities.**A New Paradigm for Representation Robustness:** The GAN-enhanced DLFR module reframes the problem of missing data from one of simple reconstruction to one of distributional consistency. By training the model to produce representations from incomplete data that are indistinguishable from those of complete data, it learns features that are inherently more robust. This adversarial approach offers a more principled way to handle data imperfection than traditional imputation or reconstruction-focused methods.

### 5.4. Limitations and Future Work

Despite its strong performance, DAST-GAN has several limitations that present avenues for future research.

**Computational Cost:** The inclusion of a GAN-based training regimen, while effective, increases the computational overhead compared to simpler models. Future work could explore more lightweight adversarial learning techniques, such as knowledge distillation from the discriminator or more efficient generator-discriminator architectures inspired by recent advances in generative modeling [50].**Generalization to In-the-Wild Data:** While tested on standard benchmarks [44,45,46], the model’s performance on truly “in-the-wild” data—encompassing a wider range of languages, cultural expressions, and environmental conditions—remains to be validated. Future research should focus on cross-dataset and cross-cultural evaluation to assess its real-world generalizability.**Interpretability:** The attention visualizations provide some insight into the model’s decision-making process. However, the complex interactions within the deep network remain partially opaque. Developing more advanced interpretability methods to explain *why* the model arrives at a particular sentiment prediction is a critical next step for building trust and facilitating error analysis.

## 6. Conclusions

This paper introduced DAST-GAN, a novel framework for robust multimodal sentiment analysis. The core contribution lies in a synergistic architecture that addresses two critical weaknesses in recent models: static fusion and representation brittleness. By integrating a Dynamic Attention Module (DAM) for adaptive fusion, a unified Spatio-Temporal Attention (STA) mechanism for holistic interaction modeling, and a novel GAN-based adversarial objective for robustness, DAST-GAN advances the state of the art.

The main findings of this study can be summarized as follows: First, DAST-GAN achieves substantial performance improvements across three benchmark datasets, reducing Mean Absolute Error (MAE) by 2.6% on CMU-MOSI (complete modality, compared to reproduced EMT baseline) and 3.76% on CMU-MOSEI (incomplete modality, compared to literature-reported EMT-DLFR ^‡^) compared to strong baseline methods. Second, the ablation studies reveal that the three core components (DAM, STA, and GAN) exhibit strong synergistic effects, with their combined contribution exceeding the sum of individual improvements. The t-SNE visualization analysis shows improved feature clustering quality, with tighter intra-class compactness and clearer inter-class boundaries compared to baseline methods, which contributes to enhanced classification accuracy. Third, the framework maintains robust performance under varying degrees of modality corruption, with graceful degradation from MAE of 0.728 (10% missing) to 1.558 (90% missing), consistently outperforming baselines across all missing rates and patterns. Fourth, visualization analyses confirm that the Dynamic Attention Module successfully adapts to context-dependent modality importance, allocating significantly different attention weights across emotional contexts (e.g., 52.1% text attention for negative emotions vs. 36.8% audio attention for positive emotions).

While the current work demonstrates strong performance across benchmark datasets, several avenues for future research remain. As detailed in Section 5.4, key priorities include improving computational efficiency, validating generalization on diverse in-the-wild data, and enhancing model interpretability. Beyond these, integrating external knowledge sources and exploring synergies with emerging foundation models could further advance robust multimodal sentiment understanding in real-world applications.

## Figures and Tables

**Figure 1 sensors-25-07109-f001:**
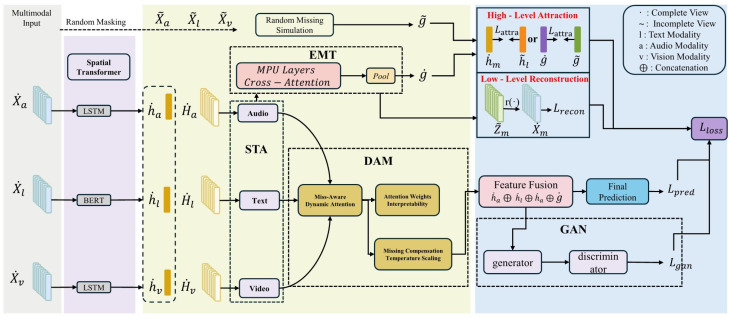
The architecture of DAST-GAN, illustrating its three core innovations. **Dynamic Attention Module (DAM):** Utilizes global semantic context to generate dynamic weights ($\alpha_{m}$), enabling adaptive fusion of unimodal features. **Spatio-Temporal Attention (STA):** A unified module that simultaneously models temporal dependencies within each modality and spatial (cross-modal) interactions between them. **GAN-Enhanced Dual-Level Feature Restoration (DLFR):** A novel adversarial framework where the main model acts as a generator (*G*) to produce robust representations from incomplete data ($\tilde{X}$) that a discriminator (*D*) cannot distinguish from representations derived from complete data (*X*). This is complemented by high-level semantic attraction and low-level feature reconstruction losses to ensure comprehensive robustness.

**Figure 2 sensors-25-07109-f002:**
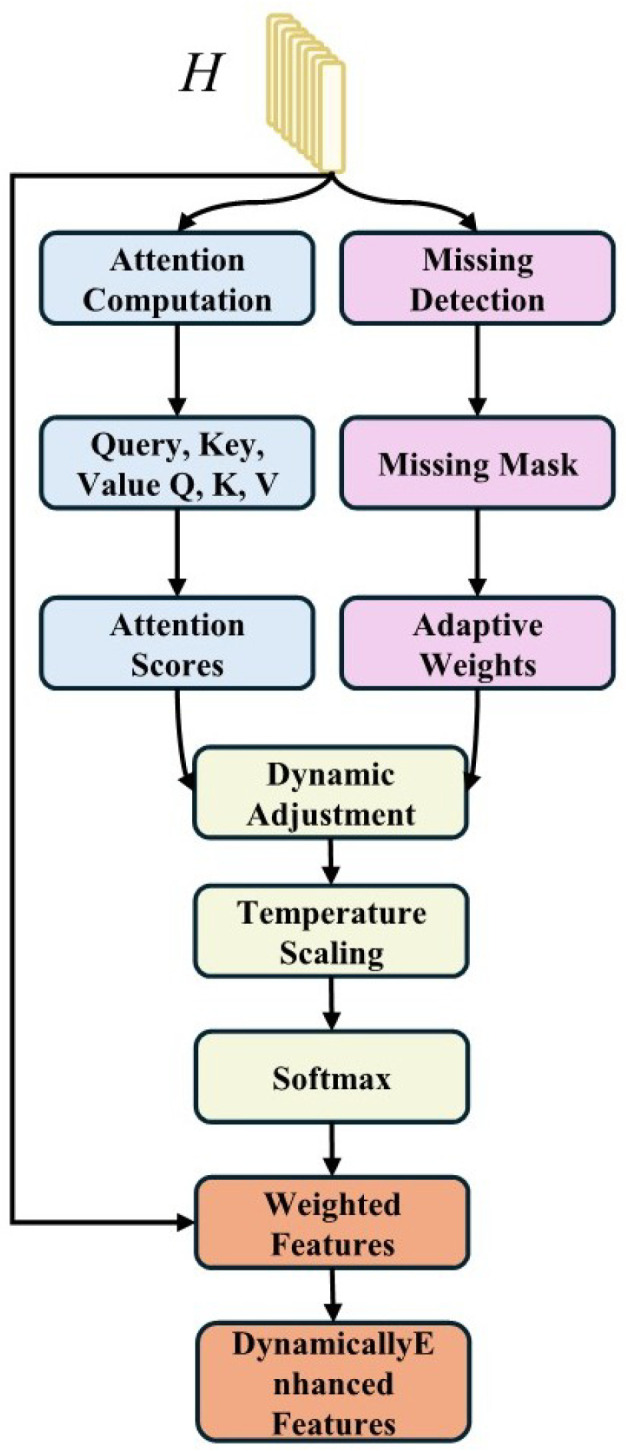
Dynamic Attention Module (DAM) architecture.

**Figure 3 sensors-25-07109-f003:**
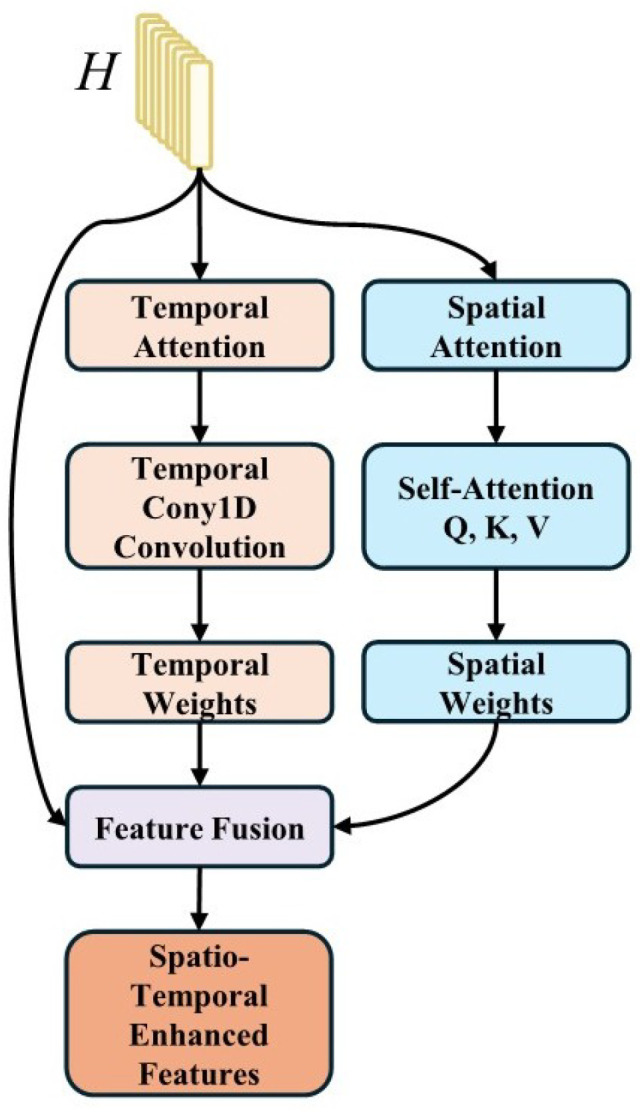
Spatio-Temporal Attention (STA) mechanism.

**Figure 4 sensors-25-07109-f004:**
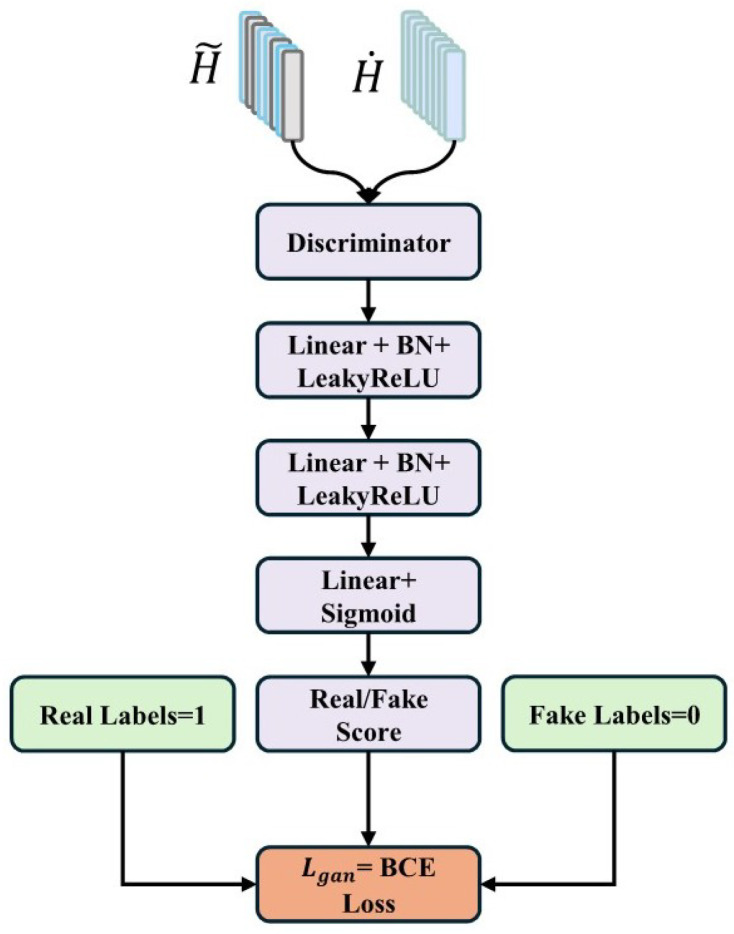
GAN-based adversarial training framework.

**Figure 5 sensors-25-07109-f005:**
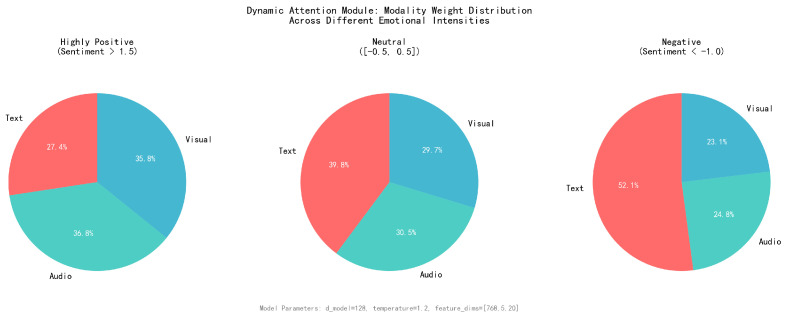
Dynamic Attention Module: Modality Weight Distribution Across Different Emotional Intensities.

**Figure 6 sensors-25-07109-f006:**
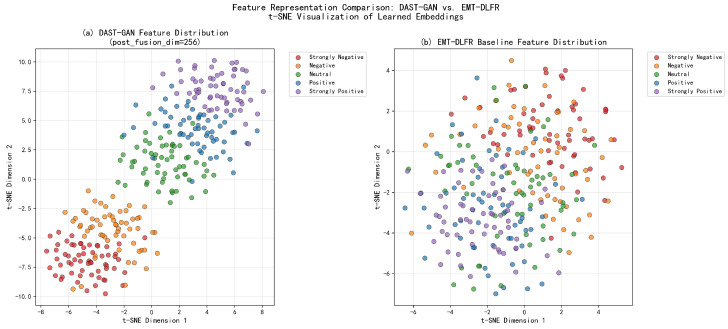
Feature Representation Comparison: DAST-GAN vs. EMT-DLFR - t-SNE Visualization of Learned Embeddings.

**Figure 7 sensors-25-07109-f007:**
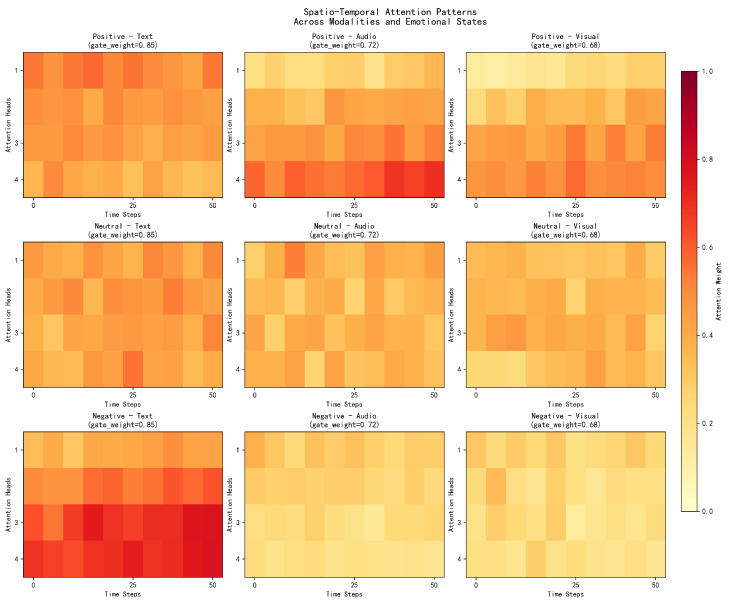
Spatio-Temporal Attention Patterns Across Modalities and Emotional States.

**Table 1 sensors-25-07109-t001:** Statistics of the three MSA benchmark datasets. The three numbers represent the number of samples with negative (<0), neutral (=0), and positive (>0) emotions, respectively.

Dataset	Train	Val	Test	Total
CMU-MOSI	552/53/679	92/13/124	379/30/277	2199
CMU-MOSEI	4738/3540/8048	506/433/932	1350/1025/2284	23,453
CH-SIMS	742/207/419	248/69/139	248/69/140	2281

**Table 2 sensors-25-07109-t002:** Detailed configuration of hyperparameters for three MSA datasets.

Hyperparameters	CMU-MOSI	CH-SIMS	CMU-MOSEI
Batch Size	32	16	32
Learning Rate	$1\times 10^{-3}$	$1\times 10^{-4}$	$1\times 10^{-3}$
BERT Learning Rate	$5\times 10^{-5}$	$2\times 10^{-5}$	$2\times 10^{-5}$
Optimizer	Adam	Adam	Adam
Early Stopping (Epochs)	8	8	8
Gradient Accumulation (Batches)	4	4	4
Hidden Unit Size in EMT	128	128	32
Number of Stacked Layers in EMT	3	2	4
Number of Attention Heads	4	4	4
Embedding Dropout	0.0	0.0	0.0
Attention Dropout	0.3	0.0	0.0
Loss Weight $\lambda_{\mathrm{gan}}$	1.0	1.0	0.5
Loss Weight $\lambda_{\mathrm{recon}}$	1.0	1.0	0.5
Loss Weight $\lambda_{\mathrm{attra}}$	0.1	0.1	0.08

**Table 3 sensors-25-07109-t003:** Performance comparison on CMU-MOSI and CMU-MOSEI under the complete modality setting. ^†^: Results from [47]. ^‡^: Results from [6]. All other results are reproduced under the same settings using publicly available open-source code and original hyperparameters. Except for MAE, higher values of all metrics indicate better performance.

Models	CMU-MOSI						CMU-MOSEI					
	MAE ↓	Corr ↑	Acc-7 ↑	Acc-5 ↑	Acc-2 ↑	F1 ↑	MAE ↓	Corr ↑	Acc-7 ↑	Acc-5 ↑	Acc-2 ↑	F1 ↑
${\mathrm{MISA}}^{\;†}$	0.804	0.764	-	-	80.8/82.1	80.8/82.0	0.568	0.724	-	-	82.6/84.2	82.7/84.0
${\mathrm{MulT}}^{\;‡}$	0.846	0.725	40.4	46.7	81.7/83.4	81.9/83.5	0.564	0.731	52.6	54.1	80.5/83.5	80.9/83.6
Self-MM ^‡^	0.717	0.793	46.4	52.8	82.9/84.6	82.8/84.6	0.533	0.766	53.6	55.4	82.4/85.0	82.8/85.0
${\mathrm{MMIM}}^{\;‡}$	0.712	0.790	46.9	53.0	83.3/85.3	83.4/85.4	0.536	0.764	53.2	55.0	82.5/85.0	82.4/85.1
TFR-Net ^‡^	0.721	0.789	46.1	53.2	82.7/84.0	82.7/84.0	0.551	0.756	52.3	54.3	81.8/83.5	81.6/83.8
${\mathrm{EMT}}^{\;‡}$	0.705	0.798	47.4	54.1	83.3/85.0	83.2/85.0	0.527	0.774	54.5	56.3	83.4/86.0	83.7/86.0
Self-MM	0.720	0.790	46.6	53.0	83.0/84.7	82.9/84.8	0.530	0.769	53.2	55.3	82.3/85.0	82.4/85.1
EMT	0.717	0.788	47.1	53.7	82.9/84.4	82.9/84.5	0.537	0.767	53.2	55.1	82.9/85.3	83.0/85.5
**DAST-GAN**	**0.698**	**0.800**	**48.6**	**54.9**	**83.2/85.1**	**83.1/85.0**	**0.528**	**0.780**	**54.0**	**56.0**	**83.1/85.1**	**83.2/85.3**

**Table 4 sensors-25-07109-t004:** Performance comparison on CH-SIMS under the complete modality setting. ^†^: Results from [47]. ^‡^: Results from [6]. All other results are reproduced under the same settings using publicly available open-source code and original hyperparameters. Except for MAE, higher values of all metrics indicate better performance.

Models	CH-SIMS					
	MAE ↓	Corr ↑	Acc-5 ↑	Acc-3 ↑	Acc-2 ↑	F1 ↑
${\mathrm{MISA}}^{\;†}$	0.447	0.563	-	-	76.5	76.6
${\mathrm{MulT}}^{\;‡}$	0.442	0.581	40.0	65.7	78.2	78.5
Self-MM ^‡^	0.411	0.601	43.1	66.1	78.6	78.6
${\mathrm{MMIM}}^{\;‡}$	0.422	0.597	42.0	65.5	78.3	78.2
TFR-Net ^‡^	0.437	0.583	41.2	64.2	78.0	78.1
${\mathrm{EMT}}^{\;‡}$	0.396	0.623	43.5	67.4	80.1	80.1
Self-MM	0.419	0.598	42.7	65.4	78.1	78.3
EMT	0.412	0.600	42.9	65.5	79.0	79.2
**DAST-GAN**	**0.398**	**0.640**	**43.6**	**67.5**	**80.5**	**80.5**

**Table 5 sensors-25-07109-t005:** Performance comparison on CMU-MOSI and CMU-MOSEI under the incomplete modality setting. The reported results are the AUILC evaluation metrics with missing rates {0.1, 0.2, …, 1.0}. ^†^: Results from [47]. ^‡^: Results from [6]. All other results are reproduced under the same settings using publicly available open-source code and original hyperparameters. Except for MAE, higher values of all metrics indicate better performance.

Models	CMU-MOSI						CMU-MOSEI					
	MAE ↓	Corr ↑	Acc-7 ↑	Acc-5 ↑	Acc-2 ↑	F1 ↑	MAE ↓	Corr ↑	Acc-7 ↑	Acc-5 ↑	Acc-2 ↑	F1 ↑
${\mathrm{MISA}}^{\;†}$	1.202	0.405	25.7	27.4	63.9/63.7	59.0/58.8	0.698	0.514	45.1	45.7	75.2/75.7	74.4/74.0
${\mathrm{MulT}}^{\;‡}$	1.263	0.348	23.1	24.6	63.1/63.2	60.7/61.0	0.700	0.504	46.3	46.8	74.4/75.1	72.9/72.6
Self-MM ^‡^	1.162	0.444	27.8	30.3	66.9/67.5	65.4/66.2	0.685	0.507	46.7	47.3	75.1/75.4	73.7/72.9
${\mathrm{MMIM}}^{\;‡}$	1.168	0.450	27.0	29.4	66.8/66.9	64.6/65.8	0.687	0.520	46.5	47.1	75.0/75.3	73.8/74.2
TFR-Net ^‡^	1.156	0.452	27.5	30.5	67.6/67.8	65.7/66.1	0.689	0.511	46.9	47.3	74.7/74.2	72.5/73.4
EMT-DLFR ^‡^	1.106	0.486	32.5	35.6	69.6/70.3	69.6/70.3	0.665	0.546	47.9	48.8	76.4/76.9	75.2/75.9
Self-MM	1.215	0.440	28.1	30.5	66.6/67.7	66.8/67.9	0.695	0.496	46.8	47.1	75.0/75.3	74.4/75.2
EMT-DLFR	1.113	0.473	32.0	35.3	69.3/70.0	69.3/70.0	0.673	0.524	47.1	48.8	75.5/76.7	74.3/75.2
**DAST-GAN**	**1.108**	**0.490**	**33.1**	**36.0**	**69.7/70.4**	**69.6/70.4**	**0.640**	**0.563**	**48.0**	**50.2**	**77.1/77.9**	**76.2/77.2**

**Table 6 sensors-25-07109-t006:** Performance comparison on CH-SIMS under the incomplete modality setting. The reported results are the AUILC evaluation metrics with missing rates {0.1, 0.2, …, 0.5}. ^†^: Results from [47]. ^‡^: Results from [6]. All other results are reproduced under the same settings using publicly available open-source code and original hyperparameters. Except for MAE, higher values of all metrics indicate better performance.

Models	CH-SIMS					
	MAE ↓	Corr ↑	Acc-5 ↑	Acc-3 ↑	Acc-2 ↑	F1 ↑
${\mathrm{MISA}}^{\;†}$	0.293	0.053	10.6	26.4	34.8	28.9
${\mathrm{MulT}}^{\;‡}$	0.242	0.233	16.7	30.0	37.3	36.8
Self-MM ^‡^	0.231	0.258	18.3	30.5	37.4	37.5
${\mathrm{MMIM}}^{\;‡}$	0.244	0.238	17.7	29.8	37.0	36.2
TFR-Net ^‡^	0.237	0.253	17.8	30.0	37.3	37.2
EMT-DLFR ^‡^	0.215	0.287	20.4	31.9	38.4	38.5
Self-MM	0.237	0.250	19.1	30.6	36.6	36.4
EMT-DLFR	0.230	0.268	19.5	31.2	37.1	37.3
**DAST-GAN**	**0.213**	**0.290**	**20.8**	**32.3**	**38.7**	**38.7**

**Table 7 sensors-25-07109-t007:** Ablation study on CMU-MOSI dataset.

Models	CMU-MOSI					
	MAE ↓	Corr ↑	Acc-7 ↑	Acc-5 ↑	Acc-2 ↑	F1 ↑
Baseline (EMT-DLFR)	0.717	0.788	47.1	53.7	82.9/84.4	82.9/84.5
w/o DAM + STA	0.712	0.791	47.3	53.9	83.0/84.5	83.0/84.6
w/o STA + GAN	0.709	0.793	47.6	54.1	83.1/84.7	83.1/84.7
w/o DAM + GAN	0.706	0.795	47.8	54.3	83.2/84.8	83.2/84.8
w/o GAN	0.704	0.796	48.0	54.5	83.0/84.8	83.0/84.8
w/o STA	0.702	0.797	48.2	54.6	83.1/84.9	83.1/84.9
w/o DAM	0.705	0.794	47.9	54.4	83.0/84.7	83.0/84.7
**DAST-GAN (Full)**	**0.698**	**0.800**	**48.6**	**54.9**	**83.2/85.1**	**83.1/85.0**

**Table 8 sensors-25-07109-t008:** Performance under different modality missing patterns on CMU-MOSI dataset (MAE values).

Missing Pattern	10%	30%	50%	70%	90%
Random Missing	0.728	0.768	0.885	1.185	1.558
Burst Missing	0.735	0.782	0.906	1.208	1.581
Modality-wise Missing	0.742	0.798	0.927	1.231	1.604
EMT-DLFR (Random)	0.741	0.798	0.923	1.235	1.612

## Data Availability

The datasets used in this study (MOSI, MOSEI, and SIMS) are publicly available and can be accessed through the MMSA framework at https://github.com/thuiar/MMSA (accessed on 19 November 2025). No new datasets were generated during this study.

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
