# Peer review of "DAST-GAN: An Adversarial Learning Multimodal Sentiment Analysis Model Based on Dynamic Attention and Spatio-Temporal Fusion"

_sensors, 2025, doi:10.3390/s25237109_

Round 1

Reviewer 1 Report

Comments and Suggestions for Authors

This work employs an inter- and intra-modal attention combined with a GAN to address the issue of incomplete data in MSA areas, which is interesting. The methodology and experiments are well-developed. One concern is the naming of the 'spatial and temporal' attention, which is somewhat confusing at the beginning of the reading. It would be better to clarify/explain the intention of using spatial and temporal features within the context of the inter- and intra-modal framework, as in this work the spatial and temporal context is limited by the data used.

Comments on the Quality of English Language

N/A

Author Response

Comments 1: This work employs an inter- and intra-modal attention combined with a GAN to address the issue of incomplete data in MSA areas, which is interesting. The methodology and experiments are well-developed. One concern is the naming of the 'spatial and temporal' attention, which is somewhat confusing at the beginning of the reading. It would be better to clarify/explain the intention of using spatial and temporal features within the context of the inter- and intra-modal framework, as in this work the spatial and temporal context is limited by the data used.

Response 1: Thank you for pointing this out. We agree with this comment. Therefore, we have added clarifications throughout the manuscript to explain the specific meaning of "spatio-temporal attention" in the context of multimodal sentiment analysis. The changes can be found in the following locations:

We have added explanation in Section 1.2. The updated text is: "In the context of multimodal sentiment analysis, the term "spatio-temporal" has specific meanings: "temporal" refers to the sequential dependencies within each modality..." (Page 3, Lines 85-89)

We have added clarification in Section 3.3.2. The updated text is: "Cross-modal feature correlations are modeled using learnable..." (Page 8, Lines 312-314)

We have added unified explanation in Section 3.6.5. The updated text is: "This unified spatio-temporal attention mechanism (as defined in Section 1.2) simultaneously captures both temporal dependencies within modalities and spatial correlations across modalities..." (Page 12, Lines 429-432)

Reviewer 2 Report

Comments and Suggestions for Authors

A lot of effort and work have gone into this. We can see a strong adapted method that joins different models.

However, the work can be improved.

The motivations and benefits of the work should be presented more clearly. 

The applications of this model should be graphically represented in the paper.

The related work has not been investigated properly. A systematic study of the literature of this work should be presented to define the gaps in this field of research.

The selected datasets and the performance of the proposed model on these datasets should be investigated more deeply and clearly.

More justifications and explanations of the results have to be presented.

The conclusion should summarize the main findings and the directions towards future studies in this field.

Author Response

Comments : A lot of effort and work have gone into this. We can see a strong adapted method that joins different models.

However, the work can be improved.

The motivations and benefits of the work should be presented more clearly.

The applications of this model should be graphically represented in the paper.

The related work has not been investigated properly. A systematic study of the literature of this work should be presented to define the gaps in this field of research.

The selected datasets and the performance of the proposed model on these datasets should be investigated more deeply and clearly.

More justifications and explanations of the results have to be presented.

The conclusion should summarize the main findings and the directions towards future studies in this field.

Response: Thank you for the constructive feedback and the recognition of the effort put into this work. We appreciate the detailed suggestions for improvement. In response to your comments, we have made several revisions to the manuscript to enhance its clarity, depth, and impact.

1.The motivations and benefits of the work should be presented more clearly:

Added explicit research motivation in the introduction opening: "This work is motivated by the critical gap between the theoretical performance of existing MSA models and their practical applicability..." (Page 1, Lines 31-34)

Emphasized the limitations of static fusion mechanisms after Challenge 1 description: "The inability to adaptively weight modalities based on contextual cues represents a fundamental limitation..." (Page 2, Lines 51-54)

Clarified the practical need for handling partial modality loss after Challenge 2 description: "This gap between theoretical robustness and practical applicability highlights the urgent need..." (Page 2, Lines 65-68)

2.The applications of this model should be graphically represented in the paper:

We have added structured, step-by-step descriptions of how DAST-GAN functions in three representative application scenarios (Section 5.1, Page 27, Lines 944-977):

(1) Empathetic HCI (input reception → DAM redistribution → prediction),

(2) Mental Health Monitoring (DLFR recovery → DAM identification → STA analysis)

(3) Market Research (encoding → weighting → modeling → classification).

3.The related work has not been investigated properly. A systematic study of the literature of this work should be presented to define the gaps in this field of research.

We have substantially revised the Related Work section to provide a more systematic literature review and clearly identify research gaps. The updated text is: "This section provides a comprehensive and systematic review of the multimodal sentiment analysis (MSA) literature, organized along two primary dimensions: (1) the evolution of fusion strategies for complete modalities..." (Page 3, Lines 115-119)

We have added systematic review structure in Section 2.1 and Section 2.2. The updated text is: "The evolution of fusion strategies in MSA can be broadly categorized into three distinct generations..."(Page 3, Lines 121-125 and Page 4, Lines 134-137, Lines 144-148, Lines 155-159, Lines 163-167) and "Handling missing or corrupted data represents a critical challenge for real-world MSA deployment..." (Page 4, Lines 169-175 and Page 5, Lines 181-184, Lines 189-194, Lines 201-205)

We have also added a new subsection "Research Gaps and Motivations" that explicitly identifies and defines three critical research gaps(Page 5, Lines 207-227):

"Based on this systematic review of the literature, we identify three critical research gaps that remain unaddressed by existing approaches:

Gap 1: Adaptive Fusion for Context-Dependent Modality Importance...

Gap 2: Efficient Cross-Modal Attention for Scalable Fusion...

Gap 3: Robust Representations for Realistic Partial Modality Loss... "

Furthermore, we have enhanced the literature review structure by organizing fusion strategies into three distinct generations, systematically categorizing approaches for incomplete modalities into three paradigms, and explicitly linking each research gap to our proposed solutions.

"These research gaps collectively motivate the design of DAST-GAN, which introduces three synergistic innovations to address each gap: (1) a Dynamic Attention Module (DAM) for context-aware adaptive fusion; (2) a unified Spatio-Temporal Attention (STA) mechanism for efficient cross-modal interaction modeling; and (3) a GAN-enhanced adversarial learning framework for learning robust representations from incomplete data..." (Page 6, Lines 228-234)

4.The selected datasets and the performance of the proposed model on these datasets should be investigated more deeply and clearly.

Dataset Selection Rationale(Page 14, Lines 469-472):"These datasets are strategically selected to provide complementary evaluation perspectives...”

CMU-MOSI justification: "Its moderate scale (2,199 samples) and fine-grained 7-point annotation make it ideal for detailed ablation studies and nuanced performance evaluation." (Page 14, Lines 481-482)

CMU-MOSEI justification: "With 10.7× more samples and dramatically higher diversity (1,000 speakers, 250 topics), this dataset rigorously tests model scalability and the GAN-enhanced framework's robustness under incomplete modality conditions." (Page 14, Lines 489-491)

CH-SIMS justification: "This dataset validates cross-lingual generalization and tests the dynamic attention mechanism's adaptability to Chinese emotional expression, where prosodic and contextual cues are more critical than explicit verbal content." (Page 14, Lines 499-501)

Dataset Characteristics Analysis (Page 14, Lines 502-506): "The three datasets exhibit complementary class distributions: CMU-MOSI shows positive bias (679 vs. 552 negative)...”

In-Depth Performance Analysis on Complete Modality Setting (Section 4.4.1, Pages 16-18)

CMU-MOSI: "The proposed DAM and STA mechanisms effectively enhance both regression and classification performance..." (Page 17, Lines 590-594)

CMU-MOSEI: "The stable performance on this 10.7× larger dataset validates the model's scalability and generalization capability." (Page 17, Lines 599-601)

CH-SIMS: "Compared to the reproduced EMT baseline (0.412 MAE, 0.600 Corr), this represents 3.4% MAE improvement and 6.7% correlation improvement..." (Page 16-17, Lines 575-581)

In-Depth Performance Analysis on Incomplete Modality Setting (Section 4.4.2, Pages 18-19)

CMU-MOSI: "The GAN-enhanced adversarial learning effectively maintains performance stability across...” (Page 18, Lines 627-629)

CMU-MOSEI: "The larger improvement margin on CMU-MOSEI (compared to 0.45% on CMU-MOSI) demonstrates that the GAN-based framework scales effectively with dataset size and diversity..." (Page 18, Lines 635-639)

CH-SIMS: "The larger improvement margin on CH-SIMS compared to English datasets (0.45% on CMU-MOSI, 1.1% on CMU-MOSEI)..." (Page 19, Lines 661-666)

 Cross-Dataset Performance Summary : ”In summary, the evaluation across three datasets under both complete and incomplete modality settings...”(Page 20 , Lines 673-678)

Dataset-Model Design Synergy Discussion:"The three datasets provide complementary evaluation perspectives: CMU-MOSI (2,199 samples) enables detailed ablation studies with its moderate scale and fine-grained annotations...”(Page 27 , Lines 979-988)

5.More justifications and explanations of the results have to be presented:

Enhanced Complete Modality Results Explanation (Section 4.4.1): ”The performance gains can be attributed to three synergistic mechanisms: DAM's context-aware...”(Page 18 , Lines 606-612)

Deepened Incomplete Modality Robustness Analysis (Section 4.4.2): “This robustness stems from the GAN discriminator enforcing distributional consistency....”(Page 19 , Lines 645-650)

Strengthened Ablation Study Interpretation (Section 4.5.2): “Notably, the synergistic effect is non-additive: removing DAM causes 0.7% degradation (0.705 vs 0.698) and removing STA causes 0.4% (0.702 vs 0.698)...”(Page 21 , Lines 717-722)

Quantified Visualization Analysis (Section 4.6.2): “The improved clustering quality (with visibly tighter intra-class compactness and clearer inter-class separation) directly contributes...”(Page 23, Lines 819-823)

Expanded Error Analysis (Section 4.6.4): ”Detailed analysis reveals that failure cases primarily fall into three categories...”(Page 25, Lines 868-875)

6.The conclusion should summarize the main findings and the directions towards future studies in this field:

Main Findings Summary:The updated text is: "The main findings of this study can be summarized as follows: First, DAST-GAN achieves substantial performance improvements across three benchmark datasets...” (Page 28-29, Lines 1029-1045)

Future Research Directions: The updated text is: "While the current work demonstrates strong performance across benchmark datasets...”(Page 29, Lines 1046-1051)

Reviewer 3 Report

Comments and Suggestions for Authors

The submitted paper presents a novel framework for multimodal sentiment analysis (MSA) that aims to address two central challenges in the field: the efficient modeling of cross-modal dependencies in unaligned data and the robustness of emotion recognition under missing or corrupted modalities. The authors introduce DAST-GAN, a model that integrates a Dynamic Attention Module (DAM), a Spatio-Temporal Attention (STA) mechanism, and a GAN-enhanced learning strategy within a unified Transformer-based architecture. The proposed framework is evaluated on three benchmark datasets—CMU-MOSI, CMU-MOSEI, and CH-SIMS—showing consistent improvements over strong baselines in both complete and incomplete modality scenarios.

The main contribution of the paper lies in the integration of dynamic and adaptive fusion strategies with adversarial learning to improve both accuracy and robustness in multimodal sentiment prediction. The DAM adaptively weights modalities at the utterance level, enabling the model to adjust the relative importance of linguistic, acoustic, and visual cues depending on context. The STA module introduces a joint treatment of temporal and spatial dependencies, which allows simultaneous modeling of intra- and inter-modal relationships rather than processing them sequentially. The GAN component strengthens the model’s resistance to missing modalities by enforcing representational consistency between complete and incomplete inputs through an adversarial objective and a dual-level restoration mechanism. Together, these elements lead to measurable gains across multiple datasets and evaluation metrics, notably reducing MAE and increasing correlation scores under both complete and missing modality settings. The paper also includes a thorough ablation study that convincingly isolates the contribution of each component.

The novelty of the paper lies primarily in the synergistic integration of these three mechanisms rather than in the individual components themselves. While dynamic attention and adversarial training have been explored previously in related contexts, their joint use in a multimodal Transformer specifically tailored for sentiment analysis represents a methodological advancement. The proposed GAN-based feature restoration is particularly noteworthy for its ability to handle partially missing modalities, a realistic and often overlooked scenario in practical MSA applications. Moreover, the spatio-temporal attention unification is a nontrivial extension of prior Transformer-based fusion strategies, offering a more coherent framework for modeling multimodal dependencies.

Overall, the manuscript is well structured and technically sound. The experimental section is comprehensive, with detailed benchmarks, hyperparameter specifications, and comparisons that enhance reproducibility. The writing is clear and the motivations are well grounded in current literature. However, there are some minor aspects that could be improved.

1 - The presentation would benefit from a more explicit and systematic discussion of computational trade-offs associated with the proposed architecture. While the paper includes a brief complexity analysis, the practical implications of integrating adversarial training into a Transformer-based multimodal model are not sufficiently explored. In particular, GAN frameworks are well known to introduce additional computational overhead and training instability due to the adversarial optimization between the generator and discriminator. A deeper analysis quantifying this cost in terms of training time, convergence behavior, and GPU resource consumption would help contextualize the performance improvements. For example, reporting the number of epochs required for convergence compared to non-adversarial baselines, or the sensitivity of the model to hyperparameter tuning in the GAN component, would clarify the efficiency–accuracy trade-off. Furthermore, an empirical discussion of how the addition of the adversarial loss influences gradient dynamics, model variance, and the stability of multimodal feature alignment could strengthen the reader’s understanding of the technical challenges behind the implementation.

2 - In addition, while the quantitative results convincingly show the robustness of DAST-GAN under varying degrees of modality incompleteness, the paper would benefit from qualitative illustrations of this robustness. Visualizations of reconstructed or inferred features for missing modalities—such as t-SNE plots, attention maps, or example frames from the video modality—would provide an intuitive view of how the model compensates for incomplete data. Presenting such qualitative analyses would also demonstrate whether the adversarially learned features preserve the semantic coherence of the original multimodal signals, an aspect that cannot be fully captured by MAE or correlation metrics alone. For instance, showing side-by-side comparisons of predicted sentiment trajectories under complete versus partially missing modalities would greatly enhance interpretability and provide evidence that the model’s robustness extends beyond numerical performance gains.

3 - Finally, the reproducibility and transparency of the study could be significantly improved through the public release of the implementation and pretrained weights. The proposed architecture involves multiple interdependent modules, the Dynamic Attention Module (DAM), Spatio-Temporal Attention (STA), and GAN-based Dual-Level Feature Restoration (DLFR), whose interactions are complex and sensitive to implementation details. Making the code and pretrained models available would enable the research community to verify the reported results, perform fair comparisons, and explore extensions or adaptations to related tasks. Open-sourcing the code would also align the paper with current standards of reproducibility and open science that are now expected in leading venues within the field of multimodal learning.

In summary, this paper makes a valuable contribution to the literature on multimodal sentiment analysis by proposing a well-motivated, methodologically integrated, and empirically validated model. Its dynamic attention and adversarial robustness framework represent a meaningful advance over static fusion methods and reconstruction-based baselines. Subject to minor revisions and the suggestion to make the code publicly available, I consider the paper suitable for publication.

Author Response

Comments: The submitted paper presents a novel framework for multimodal sentiment analysis (MSA) that aims to address two central challenges in the field: the efficient modeling of cross-modal dependencies in unaligned data and the robustness of emotion recognition under missing or corrupted modalities. The authors introduce DAST-GAN, a model that integrates a Dynamic Attention Module (DAM), a Spatio-Temporal Attention (STA) mechanism, and a GAN-enhanced learning strategy within a unified Transformer-based architecture. The proposed framework is evaluated on three benchmark datasets—CMU-MOSI, CMU-MOSEI, and CH-SIMS—showing consistent improvements over strong baselines in both complete and incomplete modality scenarios.

The main contribution of the paper lies in the integration of dynamic and adaptive fusion strategies with adversarial learning to improve both accuracy and robustness in multimodal sentiment prediction. The DAM adaptively weights modalities at the utterance level, enabling the model to adjust the relative importance of linguistic, acoustic, and visual cues depending on context. The STA module introduces a joint treatment of temporal and spatial dependencies, which allows simultaneous modeling of intra- and inter-modal relationships rather than processing them sequentially. The GAN component strengthens the model’s resistance to missing modalities by enforcing representational consistency between complete and incomplete inputs through an adversarial objective and a dual-level restoration mechanism. Together, these elements lead to measurable gains across multiple datasets and evaluation metrics, notably reducing MAE and increasing correlation scores under both complete and missing modality settings. The paper also includes a thorough ablation study that convincingly isolates the contribution of each component.

The novelty of the paper lies primarily in the synergistic integration of these three mechanisms rather than in the individual components themselves. While dynamic attention and adversarial training have been explored previously in related contexts, their joint use in a multimodal Transformer specifically tailored for sentiment analysis represents a methodological advancement. The proposed GAN-based feature restoration is particularly noteworthy for its ability to handle partially missing modalities, a realistic and often overlooked scenario in practical MSA applications. Moreover, the spatio-temporal attention unification is a nontrivial extension of prior Transformer-based fusion strategies, offering a more coherent framework for modeling multimodal dependencies.

Overall, the manuscript is well structured and technically sound. The experimental section is comprehensive, with detailed benchmarks, hyperparameter specifications, and comparisons that enhance reproducibility. The writing is clear and the motivations are well grounded in current literature. However, there are some minor aspects that could be improved.

1 - The presentation would benefit from a more explicit and systematic discussion of computational trade-offs associated with the proposed architecture. While the paper includes a brief complexity analysis, the practical implications of integrating adversarial training into a Transformer-based multimodal model are not sufficiently explored. In particular, GAN frameworks are well known to introduce additional computational overhead and training instability due to the adversarial optimization between the generator and discriminator. A deeper analysis quantifying this cost in terms of training time, convergence behavior, and GPU resource consumption would help contextualize the performance improvements. For example, reporting the number of epochs required for convergence compared to non-adversarial baselines, or the sensitivity of the model to hyperparameter tuning in the GAN component, would clarify the efficiency–accuracy trade-off. Furthermore, an empirical discussion of how the addition of the adversarial loss influences gradient dynamics, model variance, and the stability of multimodal feature alignment could strengthen the reader’s understanding of the technical challenges behind the implementation.

2 - In addition, while the quantitative results convincingly show the robustness of DAST-GAN under varying degrees of modality incompleteness, the paper would benefit from qualitative illustrations of this robustness. Visualizations of reconstructed or inferred features for missing modalities—such as t-SNE plots, attention maps, or example frames from the video modality—would provide an intuitive view of how the model compensates for incomplete data. Presenting such qualitative analyses would also demonstrate whether the adversarially learned features preserve the semantic coherence of the original multimodal signals, an aspect that cannot be fully captured by MAE or correlation metrics alone. For instance, showing side-by-side comparisons of predicted sentiment trajectories under complete versus partially missing modalities would greatly enhance interpretability and provide evidence that the model’s robustness extends beyond numerical performance gains.

3 - Finally, the reproducibility and transparency of the study could be significantly improved through the public release of the implementation and pretrained weights. The proposed architecture involves multiple interdependent modules, the Dynamic Attention Module (DAM), Spatio-Temporal Attention (STA), and GAN-based Dual-Level Feature Restoration (DLFR), whose interactions are complex and sensitive to implementation details. Making the code and pretrained models available would enable the research community to verify the reported results, perform fair comparisons, and explore extensions or adaptations to related tasks. Open-sourcing the code would also align the paper with current standards of reproducibility and open science that are now expected in leading venues within the field of multimodal learning.

In summary, this paper makes a valuable contribution to the literature on multimodal sentiment analysis by proposing a well-motivated, methodologically integrated, and empirically validated model. Its dynamic attention and adversarial robustness framework represent a meaningful advance over static fusion methods and reconstruction-based baselines. Subject to minor revisions and the suggestion to make the code publicly available, I consider the paper suitable for publication.

Response: We sincerely thank the reviewer for the thorough evaluation and constructive feedback. We are pleased that the reviewer recognizes the novelty of our synergistic integration of DAM, STA, and GAN components, as well as the comprehensive experimental validation. We have carefully addressed all three comments to strengthen the manuscript:

Response 1: 

We thank the reviewer for this insightful feedback. We have significantly expanded Section 4.7.2 (Computational Efficiency Analysis) to provide a comprehensive discussion of the computational trade-offs associated with adversarial training.(Page 26, Lines 899-924)

Convergence behavior: Specific epoch counts (35-40 epochs vs. 30-35 for baselines) and the role of multi-stage training strategy in preventing GAN instability.

GPU resource consumption: Memory usage quantification (around 8GB for batch size 32).

Training stability: Multi-seed experimental results showing low variance (standard deviation of 0.0075 MAE).

Hyperparameter sensitivity: Dataset-specific configurations demonstrating robustness without extensive tuning.

Inference efficiency: Clarification that discriminator and auxiliary modules are training-only, resulting in zero inference overhead and compact deployment.

Response 2: 

We sincerely thank the reviewer for this valuable suggestion regarding qualitative illustrations of model robustness under modality incompleteness.

We would like to clarify that Section 4.6 (Visualization and Analysis) already includes comprehensive qualitative analyses with t-SNE plots (Figure 6), attention heatmaps (Figure 7), and attention weight visualizations (Figure 5) that demonstrate the model's mechanisms underlying its robustness.

We have enhanced the descriptions in Section 4.6 with the following additions:

Section 4.6.1 (Attention Weight Visualization): We have added explanation clarifying how the visualizations demonstrate robustness mechanisms. The updated text is: "The dynamic attention mechanism revealed here explains the model's robustness under incomplete modalities...” (Page 22, Lines 766-768)

Section 4.6.2 (Feature Distribution Analysis): We have added explanation connecting the t-SNE clustering quality to robustness. The updated text is: "The tight, well-separated clusters demonstrate that the GAN-enhanced adversarial training produces semantically coherent representations...”(Page 22, Lines 795-799)

Section 4.6.3 (Temporal Attention Analysis): We have added explanation of how attention patterns enable robustness. The updated text is: "The flexible attention patterns across time and modalities demonstrate the model's capacity to redistribute focus dynamically...”(Page 24, Lines 827-830)

Response 3: 

We sincerely appreciate the reviewer's emphasis on reproducibility and transparency, which we fully recognize as essential standards in the multimodal learning community.

We would like to clarify our position regarding code availability: The implementation involves novel architectural components (DAM, STA, and GAN-based DLFR) that are subject to institutional intellectual property considerations. Due to these institutional policies, we are unable to publicly release the source code and pretrained models at this time.

However, to support reproducibility and facilitate research in this area, we commit to the following:

Detailed Implementation Description: The paper provides comprehensive technical details including architecture parameters, training procedures, hyperparameter settings, and algorithmic specifications (Sections 3 and 4.3) that should enable reimplementation by interested researchers.

Code Availability: Researchers interested in accessing the implementation can contact the corresponding author via email. We will share the code under appropriate academic use agreements on a case-by-case basis, subject to institutional approval.

Round 2

Reviewer 2 Report

Comments and Suggestions for Authors

Authors have answered all of my questions and concerns and the paper can be accepted now